# FROM CONTEXTUAL DISTRIBUTIONS TO MESSAGES: ENTROPY-GUIDED GNNS

## ABSTRACT

The Message Passing Neural Networks (MPNNs) have emerged as the dominant framework for learning on graphs. However, their expressive power is fundamentally restricted by the 1-dimensional Weisfeiler-Lehman (1-WL) test. To further improve the expressive power of MPNNs, existing methods mainly rely on higher-order or subgraph WL tests, that usually require significantly increased memory usage and computational overhead. The aim of this paper is to address the above limitations by introducing a novel message passing paradigm, that can effectively encode and propagate the structural distribution of context around each node. Instead of directly performing the message passing on $k$-tuples or subgraphs, our method encodes and propagates structural information through a compact distributional statistic, i.e., the entropy of the node context. Furthermore, we propose a kernel-based aggregation scheme to quantify the structural distribution similarities between the contexts of different nodes. Theoretical analysis and empirical evaluations indicate that the proposed framework not only achieves higher expressive power but also significantly reduces computational and memory costs.

## 1 INTRODUCTION

Graph Neural Networks (GNNs) have emerged as a powerful tool for learning on graph-structured data. Among the various GNN architectures, the Message Passing Neural Network (MPNN) have become the most popular design paradigm (Gilmer et al., 2017). However, as demonstrated in the Graph Isomorphism Network (GIN) (Xu et al., 2019), the expressive power of most existing MPNNs is fundamentally bounded by the 1-dimensional Weisfeiler-Lehman (1-WL) test (Shervashidze et al., 2011), limiting the effectiveness for graph learning. To improve the performance of MPNNs, some works have proposed to perform the message passing over $k$-tuples of nodes (Morris et al., 2019; Maron et al., 2019b; Azizian & Lelarge, 2021), that is inspired by the higher-order WL tests. Although, this manner can enable the new models to effectively capture richer structural patterns than the 1-WL test, the expressive power of such models is inherently limited by the choice of $k$. Thus, the computational and memory costs of these models grow exponentially with $k$, making it impractical for large-scale graph datasets.

To overcome the above shortcomings, the subgraph-level message passing strategy has been developed, by decomposing the input graph into a collection of subgraphs and applying MPNNs at the subgraph level. Since the subgraph structure can significantly break the local symmetries (Zhang & Li, 2021; Zhao et al., 2022; Southern et al., 2025), this strategy provides an elegant way to distinguish highly symmetric graph structures (Zhang et al., 2023; Zhang & Li, 2021), enhancing the expressive power of MPNNs. Unfortunately, this strategy still suffers from two common drawbacks. First, it heavily relies on the predefined subgraph generation or sampling policies. Second, performing message passing at the subgraph level typically increases training time. Thus, developing MPNNs that are both effective and efficient remains a challenging problem.

The aim of this work is to propose a novel framework that leverages entropy-guided message passing, addressing the limitations of existing methods. Rather than directly performing the message passing on $k$-tuples or subgraphs, we propose a novel perspective by modeling the **distribution of node contexts**. These local distributions are represented through **entropy**, providing a compact yet expressive encoding of the structural patterns. By integrating the contextual distributions into the message passage mechanism, our method avoids the redundant computation and helps reduce the

computational overhead. Theoretical analysis and empirical evaluation demonstrate that the contextual distributions effectively capture the structural patterns while enhancing the expressive power. This work marks the first exploration of the relationship between expressiveness and the distribution of eigenvalues. Overall, the proposed method achieves a favorable trade-off between the expressiveness and the efficiency, and can be easily integrated into standard message passing frameworks.

Furthermore, we introduce a *kernelized variant* of the proposed framework, that maps node features into a similarity space defined over the entropy of node contexts. Instead of directly summing all entropies, which treats all neighbors equally, our method employs the kernel function to quantify the relative structural similarity between each neighbor and the central node. When node contexts exhibit significant differences, this approach normalizes the structural variations, enabling the stable and comparable message passing. The main contributions are summarized as follows.

- **A general framework for WL and MPNNs:** We introduce a message passing scheme that propagates not only the node features but also the contextual distributions.

- **The kernel-based aggregated variant:** We propose a kernel-based aggregation mechanism to capture the similarities between different structural distributions, and present an *efficient realization* that accelerates the computation through neighborhood sampling.

- **Theoretical findings:** We theoretically prove that our model is strictly more expressive than the 1-WL and 2-WL tests, and can discriminate a large number of regular graphs that 3-WL fails to distinguish.

- **Experimental results:** We show that our method achieves superior performance and higher computational efficiency on simulation and real-world datasets, while maintaining more expressive power compared to existing models.

## 2 RELATED WORKS

**Feature-enhanced MPNN.** The feature-enhanced methods aim to enrich node representations to obtain more informative and expressive embeddings. For instance, the ID-GNN (You et al., 2021) assigns a unique identity to each node, while the GCN-RNI (Abboud et al., 2021) enhances discriminative power by injecting randomly initialized features. CLIP (Dasoulas et al., 2020) disambiguates identical node features by assigning distinct colors. However, these approaches violate the assumption of permutation invariance. RP-GNN (Murphy et al., 2019) attempts to improve the generalization by enumerating all possible node orders. While theoretically appealing, this strategy incurs substantial computational overhead and scalability challenges, particularly on large graphs. There are other methods improve the expressive power by augmenting node features with additional structural signals. For example, the AM-GCN (Wang et al., 2020) incorporates $k$-nearest neighbor features, the GD-WL-based methods (You et al., 2019; Zhang et al., 2023) leverage pairwise node distances, the SEK-GIN (Yao et al., 2023) injects contextualized substructure information to enhance the expressiveness, the PEARL (Kanatsoulis et al., 2025) introduces efficient positional encoding, and the MoSE (Bao et al., 2024) proposes motif structural encoding (MoSE) as structural encoding based on counting graph homomorphisms. Further, topology-aware features, such as counts of specific substructures, have also been employed (Bouritsas et al., 2023). Our method can be viewed as a topology-aware feature enhancement strategy that preserves permutation invariance, thereby ensuring strong generalization capabilities without incurring significant computational cost.

**Subgraph-level MPNN.** Since the 1-WL test can not distinguish the non-isomorphic graphs with high symmetry, various subgraph-based method are proposed. The Nested GNN (Zhang & Li, 2021) and GNN-AK+ (Zhao et al., 2022) propose to perform message passing independently on the subgraph rooted at each node, implementing a subgraph-level GNN. Furthermore, the ESAN (Bevilacqua et al., 2022) proposes a subgraph-based GNN framework that performs message passing within each subgraph, and aggregates their representations in a permutation-invariant manner. These methods often require performing MPNN on full sets of subgraphs or rely on additional policies to select or sample subgraph collections. Thus, the OSAN (Qian et al., 2022) and the Policy-learn (Bevilacqua et al., 2024) respectively propose adaptive methods for learning sampled subgraphs, instead of relying on handcrafted subgraph selection. However, these methods face limitations in performance due to the sampling bias and additional computational overhead.

**Higher-order WL and K-hop MPNN.** Morris et al. (2020b) introduce the local variants and corresponding neural network architectures that focus on a subset of the original neighborhood. Moreover, Morris et al. (2022) focus on a subset of all $k$-tuples, specifically those that induce subgraphs with at least connected components. Inspired by the higher-order WL test, there are some methods that extend 1-hop message passing to $K$-hop message passing (Nikolentzos et al., 2020; Abu-El-Haija et al., 2019). In other words, the model performs message passing directly over the $K$-hop neighborhood. However, the expressive power of $K$-hop MPNN is still limited (Feng et al., 2022). Therefore, Abboud et al. (2022a) introduces SP-MPNN, which enables nodes to communicate through the shortest path neighborhoods, thereby facilitating richer and more meaningful representations. Furthermore, the KP-MPNN (Feng et al., 2022) is proposed to address the challenge by extracting the peripheral subgraph. It focuses on the peripheral edges within each node-rooted subgraph, as well as the peripheral subgraphs formed by these edges. However, this approach heavily relies on the peripheral subgraph extraction, potentially limiting its flexibility and generality.

## 3 PRELIMINARY CONCEPTS

**Notations.** Let $G = (V, E)$ be an undirected graph, where $V$ denotes the set of nodes and $E$ denotes the set of edges. For a node $u$, let $\mathcal{N}_u^k$ represent the set of its $k$-hop neighbors. We define the neighborhood up to $k$-hops as $\mathcal{N}_u^{1:k} = \bigcup_{i=1}^k \mathcal{N}_u^i$.

**Graph entropy.** Given a graph $G$, let $A$ be its adjacency matrix and $D$ be its degree matrix. The Laplacian matrix can be computed as $L = D - A$. Let $\{\lambda_i\}_{i=1}^{|V|}$ denote the eigenvalues of $L$, and let $\text{tr}(L)$ represent the trace of the Laplacian matrix. The von Neumann entropy of the graph $G$ is:

$$\mathcal{H}_{VN}(G) = -\sum_{i=1}^{|V|} \frac{\lambda_i}{\text{tr}(L)} \log\left(\frac{\lambda_i}{\text{tr}(L)}\right). \tag{1}$$

**Graph isomorphism.** Given two graphs $G_1 = (V_1, E_1)$ and $G_2 = (V_2, E_2)$, if there exists a bijection $f : V_1 \to V_2$ such that for every edge $(u, v) \in E_1$, there is a corresponding edge $(f(u), f(v)) \in E_2$, then $G_1$ and $G_2$ are isomorphic. Since there is no clear metric that can definitively characterize the structure of a graph, the graph isomorphism problem remains NP-hard.

**Weisfeiler-Lehman isomorphism test.** The Weisfeiler-Lehman (1-WL) test is a classic algorithm used to test graph isomorphism (Shervashidze et al., 2011). It operates by iteratively refining the node labels by aggregating the neighborhood information. Given a graph $G$, assume $l^0(u)$ represents the initial label of node $u$. In each iteration $t$, the 1-WL algorithm constructs the multi-set label $\mathcal{L}_{\mathcal{N}}^t$ for each node $u$ by aggregating the labels of its 1-hop neighborhood nodes $\mathcal{N}_u^1$, i.e., $\mathcal{L}_{\mathcal{N}}^t(u) = \{l^{t-1}(v) \mid v \in \mathcal{N}_u^1\}$. The 1-WL algorithm then merges the multi-set label $\mathcal{L}_{\mathcal{N}}^t$ with the current label of node $u$, and maps the sequence into a new label $l^t(u)$ through a Hash function:

$$l^t(u) = \text{Hash}(l^{t-1}(u), \mathcal{L}_{\mathcal{N}}^{t-1}(u)). \tag{2}$$

The 1-WL test repeats the iteration until the node labels no longer change or the condition meets the largest iteration.

**Message passing neural network (MPNN).** The MPNN framework generalizes the 1-WL test to continuous node and edge feature spaces by iteratively propagating and aggregating messages across a graph. It consists of two main components: a message passing phase and a readout phase. Let $G$ be an undirected graph, and $\boldsymbol{h}_u^t$ denote the hidden state of node $u$ at iteration $t$. In the message passing phase, the node representations are updated by aggregating the features of neighboring nodes and the associated edges. The node state at iteration $t + 1$ is computed as:

$$\boldsymbol{h}_u^{t+1} = \mathbf{U}^t(\boldsymbol{h}_u^t, \boldsymbol{m}_u^{t+1}), \quad \boldsymbol{m}_u^{t+1} = \sum_{v \in \mathcal{N}_u^1} \mathbf{M}^t(\boldsymbol{h}_u^t, \boldsymbol{h}_v^t, \boldsymbol{e}_{uv}), \tag{3}$$

where $\mathbf{M}^t$ and $\mathbf{U}^t$ denote the message and update functions at iteration $t$, and $\boldsymbol{e}_{uv}$ represents the edge feature vector between nodes $u$ and $v$. After $T$ iterations, a readout function aggregates the final node embeddings to form a graph-level embedding:

$$\boldsymbol{y} = \mathbf{R}(\{\boldsymbol{h}_u^T \mid u \in G\}), \tag{4}$$

where $\mathbf{R}$ is a permutation-invariant readout function, e.g., sum, mean, or a learned pooling function.

As a specific instantiation of MPNNs, the GIN (Xu et al., 2019) defines the message and update function as:

$$\boldsymbol{h}_u^{t+1} = \text{MLP}((1 + \epsilon^t)\boldsymbol{h}_u^t + \boldsymbol{m}_u^{t+1}), \quad \boldsymbol{m}_u^{t+1} = \sum\nolimits_{v \in \mathcal{N}_u^1} \boldsymbol{h}_v^t \tag{5}$$

where $\epsilon^t$ is a learnable parameter that controls the importance of the node's previous state.

**K-hop Message passing neural network (K-hop MPNN).** In the K-hop MPNN (Nikolentzos et al., 2020), the messages from 1-hop to K-hop neighborhoods are simultaneously propagated and aggregated, and a combination function is applied to integrate the resulting multi-hop node representations. The standard K-hop MPNN updates the node representation at iteration $t$ as:

$$\boldsymbol{h}_u^{t+1,k} = \mathbf{U}^{t,k}(\boldsymbol{h}_u^t, \boldsymbol{m}_u^{t+1,k}), \quad \boldsymbol{m}_u^{t+1,k} = \sum\nolimits_{v \in \mathcal{N}_u^k} \mathbf{M}^{t,k}(\boldsymbol{h}_u^t, \boldsymbol{h}_v^t, \boldsymbol{e}_{uv}^k), \tag{6}$$

where $v$ is the $k$-hop neighbor of the rooted node $u$, $\boldsymbol{h}_u^{t+1,k}$ denotes the representation of the rooted node $u$ at layer $t + 1$, aggregated from its $k$-hop neighborhood, and $\boldsymbol{e}_{uv}^k$ denotes the shortest path or random walk between $u$ and $v$. The final node representation at iteration $t + 1$ is updated through a combination function $\mathbf{C}$:

$$\boldsymbol{h}_u^{t+1} = \mathbf{C}^{t+1}\{\boldsymbol{h}_u^{t+1,k} | k = 1, 2, .., K\}. \tag{7}$$

## 4 THE CONTEXTUAL DISTRIBUTION MESSAGE PASSING FRAMEWORK

In this section, we introduce the Contextual Distribution (ConD)-based message passing framework. We begin by presenting the ConD-WL, a novel approach that iteratively updates node labels by aggregating the contextual distributions, enhancing the traditional WL test. This establishes the theoretical foundation for our main contribution, i.e., ConD-MPNN. The ConD-MPNN further extends this idea by propagating contextual **entropy** representations through a learnable message passing process. We then provide a theoretical analysis and demonstrate the expressive power of both variants independently.

### 4.1 THE OVERVIEW OF THE PROPOSED FRAMEWORK

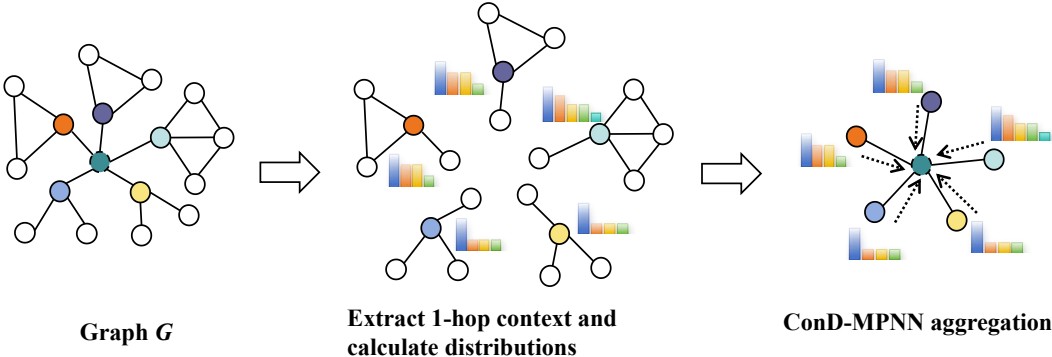

**Graph $G$**     **Extract 1-hop context and calculate distributions**     **ConD-MPNN aggregation**

Figure 1: The contextual distribution message passing mechanism. The colored histograms represent the contextual distributions associated with each neighboring node.

The proposed ConD-based framework consists of two main stages. First, we extract the context centered at each node $u$. Inspired by the context extraction approaches in (Yao et al., 2023; Abboud et al., 2022b), which construct context by considering neighboring nodes within a certain hop distance, our method directly leverages all neighbors whose distance from $u$ is between 1 to $k$, and the resulting context is formally defined in Definition 1.

**Definition 1.** *(The $k$-hop context $S_u^k$) Given a graph $G(V, E)$ and a specified center node $u \in V$, the node set $V_{S_u^k}$ of the $k$-hop context $S_u^k$ is defined as $V_{S_u^k} = \{v \in V | d(u, v) \le k\}$, where $d(u, v)$ represents the shortest path or random walk distance from node $u$ to node $v$. The edge set $E_{S_u^k}$ contains all edges between nodes in $V_{S_u^k}$, i.e., $E_{S_u^k} = \{(v_1, v_2) \in E | v_1, v_2 \in V_{S_u^k}\}$.*

The detailed algorithm for extracting the $k$-hop context $S_u^k$ is shown in the Appendix A.1. Build upon this, we can extract the local structural context of each node. In the second stage, these contexts are encoded into distributional statistics that capture their structural complexity. The resulting representations are then propagated to the neighboring nodes through a message passing mechanism. This bridges the gap between the ConD-based WL test and the learnable ConD-based MPNNs, providing a unified perspective on structural encoding. An overview of the proposed ConD-based framework is illustrated in Figure 1. In this example, we extract the 1-hop context of each neighbor, represent it by its node degree distribution, and propagate this contextual information to the center node.

## 4.2 THE CONTEXTUAL DISTRIBUTION WL TEST

We begin by introducing the Contextual Distribution Weisfeiler-Lehman (ConD-WL) algorithm. The ConD-WL method aggregates not only the labels (as in the 1-WL test) but also the distribution of the $k$-hop contexts centered at each node. Specifically, the ConD-WL constructs multisets based on the contextual distributions rooted at neighboring nodes, and combines them with label information for joint hash encoding. Thus, in each iteration $t$, the label of node $u$ is updated as:

$$l^t(u) = \text{Hash}(l^{t-1}(u), \{l^{t-1}(v) \mid v \in \mathcal{N}_u^1\}, \{\mathcal{D}_v^k \mid v \in \mathcal{N}_u^1\}), \tag{8}$$

where $\mathcal{D}_v^k$ denotes the multiset of $k$-hop context distribution centered at node $v$. We further conduct a theoretical analysis of the expressive power of ConD-WL.

**Proposition 1.** *The ConD-WL is strictly powerful than the 1-WL and 2-WL tests, i.e., if two non-isomorphic graphs can be distinguished by the 1&2-WL tests, they must be distinguished by ConD-WL, **but not vice versa**.*

The proof is shown in the Appendix A.2. The proposition indicates that the proposed method can distinguish more non-isomorphic graphs than the 1&2-WL tests.

Since the WL test relies on iteratively aggregating discrete label values, and to seamlessly incorporate distributions, we instantiate the context distribution in this paper using the degree distribution for ConD-WL, i.e., $\mathcal{D}_v^k = \{deg(j) \mid j \in V_{S_v^k}\}$. To illustrate the effectiveness of ConD-WL, we present a toy example in Figure 2. The two graphs $G_1$ and $G_2$ are indistinguishable by the 1-WL test. In contrast, the ConD-WL can capture subtle structural differences by incorporating degree distributions of node contexts. For example, the distribution multiset of the 1-hop context $S_u^1$ rooted at node $u$ in $G_1$ is $\{2, 2, 2\}$, whereas the corresponding context $S_v^1$ in $G_2$ has a distribution multiset of $\{2, 1, 1\}$. These distinct multisets are subsequently mapped to different labels via a hashing function, allowing ConD-WL to effectively distinguish between the two non-isomorphic graphs.

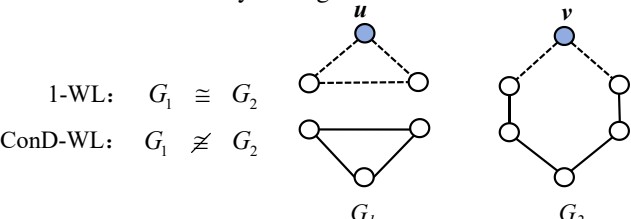

$$
\begin{array}{ll}
\text{1-WL:} & G_1 \cong G_2 \\
\text{ConD-WL:} & G_1 \ncong G_2
\end{array}
$$

Figure 2: An illustrative toy example of the ConD-WL algorithm, where $\cong$ denotes graph isomorphism and $\ncong$ denotes non-isomorphism.

## 4.3 THE ENTROPY-BASED CONTEXT ENCODING IN MPNNS

To extend the ConD-WL to differentiable graph learning, we propose the ConD-MPNN, a learnable, continuous variant of the ConD-based framework that directly integrates the contextual distribution encoding into the MPNN pipeline. Notably, the discrete degree distribution used in ConD-WL, typically represented as a multiset, has a dimensionality that scales with the number of nodes, limiting its scalability and applicability to large-scale datasets. In contrast, the ConD-MPNN effectively captures structural complexity by using the entropy, summarizing intricate structural patterns into a concise, single-dimensional representation. This significantly enhances scalability and efficiently captures structural information.

To enhance information propagation in the MPNN framework, we employ the more expressive von Neumann entropy $H_{VN}(S_u^k)$ to capture the distribution of each node context $S_u^k$. The entropy

calculation is provided in Eq.1. Therefore, the message passing phase in the ConD-MPNN can be defined as follows:

$$\boldsymbol{h}_u^{t+1} = \mathbf{U}^t(\boldsymbol{h}_u^t, \boldsymbol{m}_u^{t+1}), \quad \boldsymbol{m}_u^{t+1} = \sum\nolimits_{v \in \mathcal{N}_u^1} \mathbf{M}^t(\boldsymbol{h}_u^t, \boldsymbol{h}_v^t, \phi(\mathcal{H}_{VN}(S_u^k)), \phi(\mathcal{H}_{VN}(S_v^k)), \boldsymbol{e}_{uv}), \quad (9)$$

where $\phi$ denotes a learnable linear transformation applied to the context entropy, $\mathbf{M}^t$ and $\mathbf{U}^t$ follow the generic message passing definition in Eq. 3. We further justify our choice of von Neumann entropy over structural Shannon entropy (Shannon, 1948) through a case study in the Appendix A.3, along with a detailed comparison presented in the Appendix A.9.9.

We now provide a theoretical analysis to characterize the expressive power of ConD-MPNN. It is known that MPNNs are at most as expressive as the 1-WL test (Xu et al., 2019). Thus, the following Corollary 1 holds.

**Corollary 1.** *When MPNN is expressive as the 1-WL, the ConD-MPNN is strictly more powerful than both the 1&2-WL tests in distinguishing non-isomorphic graphs.*

To further highlight the expressive power of ConD-MPNN, we consider a canonical class of challenging graph instances, i.e., $d$-regular graphs, which are known to be indistinguishable not only by the 1&2-WL test but also by the 3-WL test. To demonstrate that the proposed method can overcome this limitation, we provide a theoretical analysis of how many distinct regular graphs ConD-MPNN can differentiate, as outlined in Theorem 1.

**Theorem 1.** *For the family $\mathcal{G}_{d,|V|}$ of all $d$-regular graphs with $|V|$ nodes, the ConD-MPNN with von Neumann entropy can distinguish at most $\mathcal{O}(|V|^2 \log^2 |V|)$ pairs of non-isomorphic regular graphs.*

The detailed procedure for constructing these non-isomorphic graphs using *edge rewiring* and the corresponding theoretical analysis, including the proof of Theorem 1, can be found in Appendix A.4. To more intuitively showcase how ConD-MPNN enhances expressive power, we present a well-known example in Appendix A.3, i.e., the Shrikhande graph and the $4 \times 4$-Rook's graph, which are indistinguishable by the 3-WL test.

In both the discrete ConD-WL and the learnable ConD-MPNN variants discussed above, the receptive field is determined by the context diameter $K$ and the layer number $t$, i.e., $max\{K, t\}$. To validate the generalizability of our framework to various backbone architectures, we extend the proposed ConD-MPNN to the $K$-hop MPNN framework (Nikolentzos et al., 2020), resulting in ConD-KMPNN. In this setting, the receptive field expands to $Kt$, enabling us to achieve the same expressive power as ConD-MPNN with significantly fewer layers. While extracting the $K$-hop neighbors in the $K$-hop MPNN framework, the corresponding $K$-hop contexts can be constructed simultaneously. This process incurs no additional computational overhead compared to standard $K$-hop MPNN. Since the expressiveness of ConD-MPNN exceeds that of traditional MPNNs, and it directly follows that the expressiveness of ConD-KMPNN is greater than that of K-MPNN. We provide a detailed discussion of this variant in the Appendix A.5.

## 4.4 THE COMPLEXITY ANALYSIS

The time complexity of our ConD-based framework primarily arises during the preprocessing phase, where $k$-hop contexts are extracted. Let $|V|$ and $|E|$ denote the number of nodes and edges in the graph, and let $d$ be the average degree of nodes. The node context extraction requires a one-time cost of $\mathcal{O}(|V| \cdot d^k)$, and the entropy computation for each context adds an additional cost of $\mathcal{O}(|V| \cdot d^{3k})$. Such preprocessing is common in the subgraph-based GNNs (Zhang & Li, 2021; Zhao et al., 2022; Huang et al., 2023) or the high-order GNNs (Nikolentzos et al., 2020; Feng et al., 2022), and is typically executed only once before training. Therefore, we primarily focus on evaluating the training time on the GPU. The message passing phase only relies on precomputed scalar dimension, and thus maintains the same time $\mathcal{O}(|V| \cdot d)$ and space complexity $\mathcal{O}(|V|)$ as standard MPNNs. We also compare the time and space complexity with the existing methods in the Appendix A.8.

## 5 THE KERNEL-BASED AGGREGATED CONTEXT ENTROPY NETWORK

Due to the significant variations in entropy scales arising from the differing distributions of local structures, directly using entropy values as features may impede the generalization of the MPNN

due to scale inconsistencies. Therefore, we propose the Kernel-based Aggregated context entropy MPNN (KerA-MPNN) to efficiently capture higher-order interactions between a node and its neighbors. Instead of directly summing all context distributions from neighboring nodes, the KerA-MPNN computes kernel values between the node context of the central node and those of its neighbors. This transforms the entropy-based context representation into a structural similarity space, allowing the model to better capture relational patterns among local structures. Inspired by the reproducing kernel proposed in (Xu et al., 2018), we define the kernel function employed in KerA-MPNN as follows.

**Definition 2.** *(The context entropy reproducing kernel) Given two contexts $S_u^k$ rooted at $u$ and $S_v^k$ rooted at $v$, the reproducing kernel of node contexts is $\mathcal{K}(S_u^k, S_v^k) = \frac{1}{2} e^{-|\mathcal{H}_{VN}(S_u^k) - \mathcal{H}_{VN}(S_v^k)|}$, where $\mathcal{H}_{VN}(S_u^k)$ denotes the the von Neumann entropy of the node context $S_u^k$.*

We provide a proof in the Appendix A.6 demonstrating that this function can effectively capture structural similarity. Due to its simplicity and positive definiteness, this kernel supports end-to-end training of our framework and provides favorable stability. The node representations in the KerA-MPNN are updated via kernel-based aggregation over entropy-encoded contexts as:

$$\boldsymbol{h}_u^{t+1} = \mathbf{U}^t(\boldsymbol{h}_u^t, \boldsymbol{m}_u^{t+1}), \quad \boldsymbol{m}_u^{t+1} = \sum_{v \in \mathcal{N}_u^1} \mathbf{M}^t(\boldsymbol{h}_u^t, \boldsymbol{h}_v^t, \mathcal{K}(S_u^k, S_v^k), \boldsymbol{e}_{uv}). \tag{10}$$

Then, we can directly derive the Proposition 2, with the detailed proof provided in Appendix A.7.

**Proposition 2.** *Let $r_u$, $r_v$ denote the entropy-based representations of the node contexts, and $z_u$, $z_v$ be their kernel-based representation. Let $\sigma_1 = |r_u - r_v|$ and $\sigma_2 = |z_u - z_v|$, then, $\sigma_2 < \sigma_1$.*

Therefore, when neighboring nodes have different node context distributions, i.e., their contextual information varies greatly, the KerA-MPNN can reduce these differences.

**Efficient realization.** To efficiently perform the kernel-based message passing, we propose an optimization strategy. Since the entropy is a measure of the complexity (Passerini & Severini, 2008), the contexts with more complex and richer structural information exhibit higher entropy. Therefore, we normalize the entropy values of the node contexts into a probability distribution and sample from it to select the most representative neighbors, i.e., those with higher entropy. We will empirically validate this in the Appendix A.9.8. This approach not only focuses on the most structurally informative regions in the local neighborhoods, but also reduces computational overhead. Thus, the message passing mechanism is updated as follows:

$$\boldsymbol{m}_u^{t+1} = \sum_{v \in \mathcal{N}_u^1} \mathbf{M}^t(\boldsymbol{h}_u^t, \boldsymbol{h}_v^t, \boldsymbol{e}_{uv}) + \phi\{\mathcal{K}(S_u^k, S_v^k)|v \sim \text{Multinomial}(p, m)\}, \tag{11}$$

where $v \sim \text{Multinomial}(p, m)$ denotes that $m$ neighbors are sampled from a multinomial distribution $p$ defined over the normalized context entropy values.

For the KerA-MPNN variant, since the number of sampled neighbors satisfies $m \ll |V|$, the additional sampling introduces negligible computational overhead. Hence, the overall time and space complexity remain $\mathcal{O}(|V| \cdot d)$ and $\mathcal{O}(|V|)$.

## 6 EXPERIMENTS

Table 1: Evaluation of expressiveness (accuracy).

| Method | EXP | CSL | SR |
|---|---|---|---|
| GCN-RNI | $98.0 \pm 1.85$ | $16.0 \pm 0.00$ | 6.67 |
| Nested GNN | $99.9 \pm 0.26$ | – | 6.67 |
| PPGN | $100.0 \pm 0.00$ | – | 6.67 |
| GIN-AK+ | $100.0 \pm 0.00$ | – | 6.67 |
| GIN | $50.0 \pm 0.00$ | $10.0 \pm 0.00$ | 6.67 |
| **ConD-GIN** | $\mathbf{100.0 \pm 0.00}$ | $\mathbf{100.0 \pm 0.00}$ | **100.0** |
| **KerA-GIN** | $\mathbf{100.0 \pm 0.00}$ | $\mathbf{100.0 \pm 0.00}$ | **100.0** |
| K-GIN | $100.0 \pm 0.00$ | $92.7 \pm 0.00$ | 6.67 |
| **ConD-KGIN** | $\mathbf{100.0 \pm 0.00}$ | $\mathbf{100.0 \pm 0.00}$ | **100.0** |
| **KerA-KGIN** | $\mathbf{100.0 \pm 0.00}$ | $\mathbf{100.0 \pm 0.00}$ | **100.0** |

In this section, we empirically evaluate the ConD-MPNN and KerA-MPNN variants in terms of expressive power, graph classification, and graph regression tasks. To better bridge theory and practice, we aim to answer the following key questions through our experiments:

**Q1:** Can we verify that the ConD-MPNN and the Ker-MPNN are more expressive than the 1-WL test and are capable of distinguishing graphs that the 3-WL test fails to distinguish? **Q2:** Can our proposed framework achieve better performance than existing methods on real-world datasets in terms of graph classification and regression tasks, and is it scalable in larger graphs? **Q3:** How does the kernelized variant alleviate the impact of context entropy distribution discrepancies while maintaining expressive power? **Q4:** Is the proposed ConD-based framework computationally efficient in terms of both runtime and memory usage?

## 6.1 DATASETS

We evaluate the expressive power of our models on four synthetic datasets specifically designed to test the limitations of MPNNs: (1) the EXP dataset (Abboud et al., 2021), consisting of 600 pairs of non-isomorphic graphs; (2) the CSL dataset (Murphy et al., 2019), comprising 150 4-regular circulant graphs; (3) the SR25 dataset (Balcilar et al., 2021), containing 15 non-isomorphic strongly regular graphs; (4) the BREC dataset (Wang & Zhang, 2024). Notably, all of these datasets cannot be distinguished by the 1-WL test, especially BREC, which remains indistinguishable even under the more powerful 3-WL and 4-WL tests. For real-world benchmarks, we evaluate the proposed ConD-MPNN and KerA-MPNN models on seven TU datasets (Morris et al., 2020a) for graph classification, and on QM9 (Ramakrishnan et al., 2014) and the ZINC (Dwivedi et al., 2023) for graph regression. To test the scalability on larger graphs, we also evaluate performance using the Peptides dataset (Dwivedi et al., 2022). The details of these datasets and the experimental settings are reported in the Appendix A.9.1 and Appendix A.9.2.

Table 2: Performance comparison of different methods on the BREC dataset.

| BREC | Basic (60) | Regular (140) | Extension graph (100) | CFI (100) | Total (400) | Accuracy |
|---|---|---|---|---|---|---|
| $I^2$-GNN | 60 | 100 | **100** | 21 | 281 | 70.2% |
| GNN-AK+ | 60 | 50 | 97 | 15 | 222 | 55.5% |
| PPGN | 60 | 50 | 100 | 23 | 233 | 58.2% |
| NGNN | 59 | 48 | 59 | 0 | 166 | 41.5% |
| GRIT | 60 | 50 | 100 | 16 | 226 | 56.5% |
| $N^2$GNN | 60 | 100 | 100 | 27 | 287 | 71.8% |
| DSS-GNN | 58 | 48 | **100** | 15 | 221 | 55.2% |
| K-GIN | 60 | 48 | 100 | 27 | 235 | 58.8% |
| KP-GIN | 60 | 106 | 98 | 11 | 275 | 68.8% |
| **ConD-KGIN (ours)** | **60** | 100 | 98 | **78** | **336** | **84%** |
| **KerA-KGIN (ours)** | **60** | **118** | 81 | 72 | 331 | 82.75% |

## 6.2 EVALUATION ON EXPRESSIVE POWER

To answer **Q1**, we conduct experiments based on the GIN (Xu et al., 2019) and K-GIN (Nikolentzos et al., 2020) backbones to evaluate the expressive power of our proposed ConD-based and KerA-based methods. We compare our models with existing powerful baselines in Table 1 and Table 2, including the GCN-RNI (Abboud et al., 2021), the Nested GNN (Zhang & Li, 2021), the PPGN (Maron et al., 2019a), the $I^2$-GNN (Huang et al., 2023), the DSS-GNN (Bevilacqua et al., 2022) and the GIN-AK+ (Zhao et al., 2022). We observe that our models achieve 100% accuracy across the EXP, CSL, and SR datasets, and further achieve the state-of-the-art performance on the BREC dataset. This demonstrates that both ConD-based and KerA-based MPNNs can distinguish all non-isomorphic graphs that cannot be distinguished by the 1-WL, and even some non-isomorphic graphs that fail to be distinguished by the 3-WL and 4-WL tests.

## 6.3 EVALUATION ON GRAPH CLASSIFICATION AND REGRESSION TASKS

**Results on TU datasets.** To answer **Q2**, we conduct graph classification experiments on the TU datasets as Table 3. We evaluate our framework on several representative backbones, including the WL (Shervashidze et al., 2011), the GIN (Xu et al., 2019), the K-GIN (Nikolentzos et al., 2020), and other backbones in the Appendix A.9.3. Since the WL test is inherently limited to discrete features, it is sufficient to compare our ConD-WL with the original WL kernel, in line with the theoretical setting in Section 4.2. As shown in Table 3, our proposed methods outperform the original backbones across almost all datasets. Notably, our models built on the K-GIN backbone outperform several state-of-the-art expressive GNNs, with ConD-KGIN achieving the best performance on six datasets. We can also observe that the KerA-GIN and KerA-KGIN performs even better than the ConD-based

Table 3: Evaluation on TuDatasets: The best and the second-best results are highlighted in **bold** and underlined, and results exceeding the baseline are shaded in gray. "OOM" denotes out-of-memory.

| Method | MUTAG | PTC | PROTEINS | IMDB-BINARY | IMDB-MULTI | DD | COLLAB |
|---|---|---|---|---|---|---|---|
| WL | 82.9 ± 0.6 | 56.8 ± 0.6 | 73.5 ± 0.4 | 71.9 ± 0.8 | 49.5 ± 0.5 | 78.6 ± 0.2 | 77.4 ± 0.4 |
| **ConD-WL** | 84.8 ± 0.6 | 57.9 ± 0.5 | 75.4 ± 0.4 | 72.7 ± 0.2 | 50.6 ± 0.2 | 78.4 ± 0.2 | 77.8 ± 0.1 |
| GIN | 89.4 ± 5.6 | 64.6 ± 7.0 | 75.9 ± 2.8 | 75.1 ± 5.1 | 52.3 ± 2.8 | 75.3 ± 2.9 | 80.2 ± 1.9 |
| **ConD-GIN** | 89.9 ± 5.0 | 66.0 ± 6.5 | 76.0 ± 3.9 | 76.3 ± 4.3 | 53.2 ± 3.4 | 78.9 ± 3.5 | 80.5 ± 1.7 |
| **KerA-GIN** | 91.5 ± 5.4 | 66.9 ± 7.5 | 76.4 ± 4.4 | 75.3 ± 2.7 | 53.1 ± 3.6 | 79.4 ± 4.4 | 80.6 ± 2.1 |
| IGN | 83.9 ± 13.0 | 58.5 ± 6.9 | 76.6 ± 5.5 | 72.0 ± 5.5 | 48.7 ± 3.4 | N/A | 78.3±2.5 |
| GNN-AK+ | 91.3 ± 7.0 | 67.7 ± 8.8 | 76.5 ± 2.5 | 75.6 ± 3.7 | 50.4 ± 3.2 | OOM | OOM |
| Nestd GNN | 87.9 ± 8.2 | 54.1 ± 7.7 | 73.9 ± 5.1 | 71.4 ± 5.9 | N/A | 77.8 ± 3.9 | N/A |
| GSN | 92.2 ± 7.5 | 67.4 ± 5.7 | 74.6 ± 5.0 | 76.8 ± 2.0 | 52.6 ± 3.6 | N/A | 82.7±1.5 |
| KP-GIN | 92.2 ± 6.5 | 66.8 ± 6.8 | 75.8 ± 4.6 | 76.6 ± 4.2 | 52.3 ± 4.1 | 80.0 ± 5.3 | N/A |
| K-GIN | 88.9 ± 8.3 | 63.8 ± 8.8 | 76.0 ± 5.0 | 73.0 ± 3.5 | 52.7 ± 3.7 | 79.5 ± 6.6 | 80.8 ± 2.1 |
| **ConD-KGIN** | **93.9 ± 5.5** | 67.4 ± 9.4 | **77.0 ± 3.1** | **76.8 ± 1.9** | **53.3 ± 2.5** | 79.7 ± 3.0 | **82.7 ± 1.2** |
| **KerA-KGIN** | **92.8 ± 4.6** | 67.6 ± 6.4 | **76.7 ± 6.5** | 76.6 ± 4.1 | **53.1 ± 3.2** | 80.4 ± 4.9 | 82.4 ± 2.1 |

models on several datasets, such as DD. This is because the DD dataset has a high edge density, which increases the likelihood of encountering neighbors with substantially different node context distributions. In contrast, the kernel-based aggregation mitigates these discrepancies by effectively normalizing the context entropy and computing distributional similarities. These findings provide strong empirical evidence for the theoretical advantages of the proposed framework.

Table 4: Comparison of GNN models on QM9 dataset (MAE).

| Target | DTNN | MPNN | PPGN | NGNN | I$^2$-GNN | SEK-GIN | K-GIN | KP-GIN | ConD-KGIN | KerA-KGIN |
|---|---|---|---|---|---|---|---|---|---|---|
| $\mu$ | 0.244 | 0.358 | **0.231** | 0.433 | 0.428 | 0.358 | 0.430 | 0.358 | 0.360 | 0.365 |
| $\alpha$ | 0.95 | 0.89 | 0.382 | 0.265 | 0.230 | 0.228 | 0.310 | 0.233 | **0.224** | 0.233 |
| $\epsilon_{HOMO}$ | 0.00388 | 0.00541 | 0.00276 | 0.00279 | 0.00261 | **0.00106** | 0.00301 | 0.00240 | 0.00237 | 0.00245 |
| $\epsilon_{LUMO}$ | 0.00512 | 0.00623 | 0.00287 | 0.00276 | 0.00267 | 0.00229 | 0.00288 | 0.00236 | **0.00228** | **0.00228** |
| $\Delta\epsilon$ | 0.0112 | 0.0066 | 0.00406 | 0.00390 | 0.00380 | 0.00335 | 0.00416 | 0.00333 | **0.00330** | 0.00346 |
| $\langle R^2 \rangle$ | 17.0 | 28.5 | 16.7 | 20.1 | 18.64 | 16.91 | 23.37 | 16.51 | **16.26** | **16.29** |
| ZPVE | 0.00172 | 0.00216 | 0.00064 | 0.00015 | 0.00014 | 0.00013 | 0.00033 | 0.00017 | 0.00018 | 0.00017 |
| $U_0$ | 2.43 | 2.05 | 0.234 | 0.205 | 0.211 | **0.0587** | 0.0602 | 0.0682 | 0.0667 | 0.0811 |
| $U$ | 2.43 | 2.00 | 0.234 | 0.200 | 0.206 | 0.0672 | 0.0504 | 0.0696 | **0.0323** | 0.0583 |
| $H$ | 2.43 | 2.02 | 0.229 | 0.249 | 0.269 | 0.073 | 0.0534 | 0.0641 | 0.0344 | **0.0268** |
| $G$ | 2.43 | 2.02 | 0.238 | 0.253 | 0.261 | 0.0592 | 0.0533 | 0.0484 | **0.0309** | 0.0389 |
| $C_\nu$ | 0.27 | 0.42 | 0.184 | 0.0811 | **0.0730** | 0.0924 | 0.1365 | 0.0869 | 0.0916 | 0.0899 |

**Results on QM9.** We further evaluate our framework on the graph regression task using the QM9 dataset. As shown in Table 4, both ConD-KGIN and KerA-KGIN demonstrate highly competitive performance across a wide range of molecular property prediction targets. Notably, the ConD-KGIN achieves the best results on 9 out of 12 targets, including $\alpha$, $\epsilon_{HOMO}$, $\epsilon_{LUMO}$, $\Delta\epsilon$, $\langle R^2 \rangle$, $U_0$, $U$, $H$, and $G$, indicating that our entropy-based context representations generalize well to complex quantum property prediction tasks. The KerA-KGIN also achieves the best performance on 5 targets, further validating the effectiveness of the proposed kernel-based strategy.

Table 5: Comparison of GNN models on ZINC dataset.

| Method | Time (s) | Params | MAE ($\downarrow$) |
|---|---|---|---|
| PPNN | >100 | 507,603 | 0.303 ± 0.068 |
| GSN | – | ~500k | 0.101 ± 0.010 |
| NGNN | – | ~500k | 0.111 ± 0.003 |
| DSS-GNN | – | **445,709** | 0.097 ± 0.006 |
| GNN-AK+ | 11.96 | ~500k | 0.091 ± 0.011 |
| Graphormer-GD | – | 502,793 | **0.081 ± 0.009** |
| K-GIN | **7.27** | 487465 | 0.097 ± 0.004 |
| KP-GIN | 7.37 | 488649 | 0.093 ± 0.007 |
| **ConD-KGIN** | 7.33 | 487465 | 0.086 ± 0.007 |
| **KerA-KGIN** | 7.69 | 506281 | 0.101 ± 0.003 |

**Results on ZINC.** We also compare our proposed methods with the existing related models on the ZINC dataset in Table 5. The ConD-KGIN achieves the second-best MAE of 0.086, and the KerA-KGIN also demonstrates competitive performance. More importantly, the ConD-KGIN outperforms the subgraph-based approaches with fewer parameters and computational efficiency, comparing to the original K-GIN backbone. These findings highlight that our framework offers a favorable trade-off between expressive power and computational efficiency.

**Results on Peptides.** To evaluate the scalability on large-scale graphs, we conduct experiments using peptide datasets in the Appendix A.9.4. We can observe from the results in Table 10 that our proposed method can still achieve the competitive performance on large-scale datasets.

## 6.4 VISUALIZATION OF CONTEXT DISTRIBUTION DISCREPANCIES

To answer **Q3**, we visualize the node context distributions of one central node and its 14 neighboring nodes in the DD dataset. As shown in the Figure 3, the left panel illustrates the distribution of entropy-based context representations, with values ranging from 1.50 to 3.25 and exhibiting considerable variation across neighboring nodes. In contrast, the right panel presents the kernel-based representations, which are mostly concentrated around 1.6. This indicates that the proposed kernel-based aggregation strategy effectively mitigates the discrepancies in node context distributions. This observation further explains why the KerA-MPNN achieves better performance than the ConD-MPNN on certain datasets, empirically validating Proposition 2. To provide a more comprehensive view, we include additional visualizations in the Appendix A.9.5, demonstrating the consistency of this phenomenon across multiple nodes..

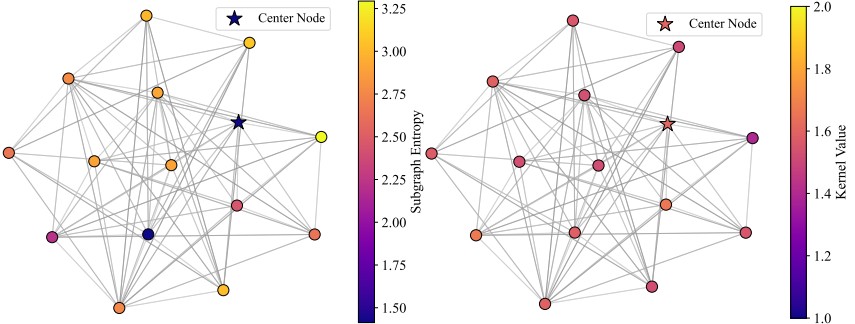

(a) Visualization on entropy value      (b) Visualization on kernel value

Figure 3: The impact of kernel aggregation based on the DD dataset.

## 6.5 ABLATION AND EFFICIENCY ANALYSIS

**Computational efficiency and scalability.** To answer **Q4**, we evaluate the end-to-end runtime including the preprocessing time, and the memory usage in Table 6. We select three datasets for the evaluations, including the dense COLLAB dataset, the large-scale DD dataset, and the sparse PROTEINS dataset. For the KP-GIN model on COLLAB dataset, the total time exceeds one day due to the preprocessing stage. The results show that our method outperforms KP-GIN (Feng et al., 2022) in runtime and demonstrates better memory efficiency than GNN-AK+ (Zhao et al., 2022) and ESAN (Bevilacqua et al., 2022).

Table 6: Efficiency comparison on three datasets."OOM" denotes out-of-memory.

| Model | DD | | COLLAB | | PROTEINS | |
|---|---|---|---|---|---|---|
| | Total time (s) | Memory | Total time (s) | Memory | Total time (s) | Memory |
| ESAN | OOM | OOM | OOM | OOM | 1645 | 2170M |
| GNN-AK+ | OOM | OOM | OOM | OOM | 1821 | 5468M |
| KP-GIN | 19096 | **1116M** | >>1 day | – | 1960 | 476M |
| **ConD-KGIN(ours)** | **7137** | 1544M | 35420 | **2620M** | **1317** | **474M** |

Further experiments are presented in the Appendix, including: (i) a study on different context size (Appendix A.9.6) (ii) a study on various context construction strategy (Appendix A.9.7); (iii) an ablation study on neigboring sampling (Appendix A.9.8); (iv) an ablation study on entropy choices (Appendix A.9.9); and (v) a comparison of kernel functions in the KerA-MPNN (Appendix A.9.10).

# 7 CONCLUSION

In this paper, we propose the Contextual Distribution Message Passing Neural Network (ConD-MPNN), a general framework designed to balance the efficiency and expressive power of MPNNs. Our approach enhances the message passing mechanism by aggregating not only the node features but also the structural distribution of their contexts, improving the expressive power in a computationally efficient manner. In addition, we introduce a Kernel-based Aggregation strategy (KerA-MPNN) to capture structural similarities between contextual distributions. Theoretical analysis and empirical results demonstrate the effectiveness and superiority of our proposed framework. Our future work will focus on improving the expressive power of the proposed model while reducing preprocessing time to achieve more efficient performance.

ETHICS STATEMENT

This work does not involve human subjects, animal experiments, or any potentially harmful insights. The datasets used in the experiments are publicly available, and all data processing steps were performed in compliance with the relevant privacy and security regulations. The authors confirm that they have read and adhered to the ICLR Code of Ethics. No conflicts of interest exist in this study, and no financial sponsorship influenced the research.

REPRODUCIBILITY STATEMENT

We have made every effort to ensure the reproducibility of our results. The code for the experiments is provided in the supplementary materials. The link is https://anonymous.4open.science/r/ConD-KGIN-F919/. The datasets used are publicly accessible, and the experimental setup is detailed in the Appendix. All theoretical claims made in the paper are supported by proofs, which are also included in the Appendix.

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

# A APPENDIX

---

**Algorithm 1** Algorithm for extracting node context based on $(1-k)$-hop neighborhood

---

**Input**: A graph $G$, adjacency matrix $A$, number of hops $k$, number of nodes $|V|$, method type (random walk or shortest path)
**Output**: The node context based on the $(1-k)$-hop neighborhood, the edge attributes for $k$-hop MPNN (the list of adjacency matrices $A_{\text{list}}$).

1: Construct an empty adjacency matrix $A_{final}$ and a list $A_{list}$ containing $k$ adjacency matrices, where the $i$-th element in the list corresponds to the adjacency matrix for the $i$-hop neighbors.
2: **for** $i \leftarrow 1$ to $k$ **do**
3:     Compute the $i$-th power of the adjacency matrix $A^{(i)} = A^i$
4:     **if** method == "random walk" **then**
5:         Update $A_{final} = A_{final} + A^{(i)}$
6:         Update $A_{list}[i] = A^{(i)}$
7:     **else if** method == "shortest path" **then**
8:         Mask nodes already existing in previous hops $A^{(i)}[A_{final} > 0] = 0$
9:         Update $A_{final} = A_{final} + A^{(i)}$
10:         Update $A_{list}[i] = A^{(i)}$
11:     **end if**
12:     **for** each node $u \in G$ **do**
13:         Extract neighbors $\mathcal{N}_u^{1:i}$ such that $A_{final}[u,v] > 0$ for $v \in \mathcal{N}_u^{1:i}$
14:         Update node context $S_u^i = A_{final}[\mathcal{N}_u^{1:i}][:,\mathcal{N}_u^{1:i}]$
15:     **end for**
16: **end for**
17: **return** the $(1-k)$-hop node context $S_u^{1:k}$, the $A_{list}$

---

## A.1 ALGORITHM FOR EXTRACTING THE NODE CONTEXT

We present the procedure for extracting the $k$-hop node context in Algorithm 1. For each node $u$, we first construct its $k$-hop neighborhood based on shortest-path or random walk distances. We then extract the corresponding induced $k$-hop node context, based on the associated node and edge sets.

## A.2 PROOF OF PROPOSITION 1

The ConD-WL enhances the standard WL test by introducing a hash function over contextual distribution sequences. This more stringent mapping intuitively offers stronger discriminative power, allowing for the distinction of a broader range of non-isomorphic substructures.

We first prove **the first part** of the proposition by contradiction. We begin by assuming, for the sake of contradiction, that there exist two non-isomorphic graphs $G_1$ and $G_2$ which can be distinguished by the 1-WL test, but are incorrectly identified as isomorphic by ConD-WL. Specifically, at the final iteration $T$, there exist two nodes $u_1 \in G_1$ and $u_2 \in G_2$ with similar but non-isomorphic local topological structures, yet the ConD-WL assigns them the same label:

$$l^T(u_1) = l^T(u_2), \tag{12}$$

and their labels are computed through the following iterative update process:

$$l^T(u) = \text{Hash}(l^{T-1}(u), \{l^{T-1}(v) \mid v \in \mathcal{N}_u^1\}, \{\mathcal{D}_v^k \mid v \in \mathcal{N}_u^1\}), \tag{13}$$

Moreover, since the hash function is injective, all three components must be identical as well, i.e.,

$$l^{T-1}(u_1) = l^{T-1}(u_2),$$

$$\text{Hash}\{l^{T-1}(v_1) \mid v_1 \in \mathcal{N}_{u_1}^1\} = \text{Hash}\{l^{T-1}(v_2) \mid v_2 \in \mathcal{N}_{u_2}^1\}, \tag{14}$$

$$\text{Hash}\{\mathcal{D}_{v_1}^k \mid v_1 \in \mathcal{N}_{u_1}^1\} = \text{Hash}\{\mathcal{D}_{v_2}^k \mid v_2 \in \mathcal{N}_{u_2}^1\}.$$

However, by assumption, the two non-isomorphic graphs can be distinguished by the 1-WL test, which implies that

$$\text{Hash}(l^{T-1}(u_1), \{l^{T-1}(v_1) \mid v_1 \in \mathcal{N}_{u_1}^1\}) \neq \text{Hash}(l^{T-1}(u_2), \{l^{T-1}(v_2) \mid v_2 \in \mathcal{N}_{u_2}^1\}) \tag{15}$$

for corresponding nodes $u$ in $G_1$ and $u_2$ in $G_2$. This contradicts the earlier conclusion that $l^T(u_1) = l^T(u_2)$ under the ConD-WL. Hence, the first part of the proposition is proven, as any pair of non-isomorphic graphs that are distinguishable by the 1-WL test can also be distinguished by ConD-WL.

Now we prove **the second part** of the proposition. We assume that there exist two nodes $u_1 \in G_1$ and $u_2 \in G_2$ at the $T-1$ iteration with the same label, and the labels of all their neighbors are also the same, i.e.,

$$l^{T-1}(u_1) = l^{T-1}(u_2), \quad l^{T-1}(v_1) = l^{T-1}(v_2), \quad \forall v_1, v_2 \in N_{u_1}^1, N_{u_2}^1. \tag{16}$$

However, the contextual distributions rooted at any neighbor $v_1$ and $v_2$ are different. Therefore, the following holds:

$$\text{Hash}(\{D_{v_1}^k \mid v_1 \in N_{u_1}^1\}) \neq \text{Hash}(\{D_{v_2}^k \mid v_2 \in N_{u_2}^1\}). \tag{17}$$

It can be directly derived that

$$\text{Hash}(l^{T-1}(u_1), \{l^{T-1}(v_1) \mid v_1 \in N_{u_1}^1\}, \{D_{v_1}^k \mid v_1 \in N_{u_1}^1\})$$
$$\neq \tag{18}$$
$$\text{Hash}(l^{T-1}(u_2), \{l^{T-1}(v_2) \mid v_2 \in N_{u_2}^1\}, \{D_{v_2}^k \mid v_2 \in N_{u_2}^1\}).$$

Thus, the ConD-WL can distinguish the two graphs, but since the two nodes $u$ and $v$ are labeled the same under 1-WL due to the identical labels and their neighbors' labels being the same, it follows that

$$\text{Hash}(l^{T-1}(u_1), \{l^{T-1}(v_1) \mid v_1 \in N_{u_1}^1\}) = \text{Hash}(l^{T-1}(u_2), \{l^{T-1}(v_2) \mid v_2 \in N_{u_2}^1\}), \tag{19}$$

which implies that 1-WL cannot distinguish these two graphs. This demonstrates that ConD-WL has strictly greater expressive power than 1-WL. Furthermore, since the expressive power of the 2-WL test has been shown to be equivalent to that of the 1-WL test (Azizian & Lelarge, 2021), it follows that ConD-WL is strictly more expressive than both the 1-WL and 2-WL tests. Therefore, proposition 1 holds.

### A.3 REASONS OF THE VON NEUMANN ENTROPY CHOICE

This section demonstrates the motivation for adopting von Neumann entropy through a case study of strongly regular graphs as Figure 4. We examine a pair of challenging instances: the $4 \times 4$ Rook's graph and the Shrikhande graph, which are indistinguishable by the 3-WL test but successfully discriminated by our method. Specifically, we denote the $4 \times 4$ Rook's graph as $G_1$ and the Shrikhande graph as $G_2$. We select one node in each graph as the center node, marking them with a star. Focusing on the starred center nodes $u \in G_1$ and $v \in G_2$, we extract their 1-hop node contexts. The edges of the node contexts are highlighted by red edges in Figure 4. Both contexts exhibit identical degree distributions, i.e., $\{5, 3, 3, 3, 3, 3\}$.

**Definition 3.** *The graph Shannon entropy is defined as:*

$$\mathcal{H}_S(G) = -\sum_{v \in V_G} \frac{deg(v)}{\sum_{m \in V_G} deg(m)} log \frac{deg(v)}{\sum_{m \in V_G} deg(m)}. \tag{20}$$

*where $deg(v)$ denotes the degree of node $v$.*

If we compute the structural Shannon entropy for degree distribution (See Definition 3), we find $H_S(S_u^1) = H_S(S_v^1)$. However, when we compute the eigenvalue distribution using the von Neumann entropy, we have $H_{VN}(S_u^1) = 1.686$ and $H_{VN}(S_v^1) = 1.698$, i.e, $H_{VN}(S_u^1) \neq H_{VN}(S_v^1)$, then our proposed ConD-MPNN can distinguish the two non-isomorphic graphs. We also conduct experiments comparing with different entropy choices in A.9.9.

### A.4 PROOF OF THEOREM 1

Due to the high symmetry and structural similarity of regular graphs, which are not isomorphic, we use edge rewiring to construct non-isomorphic graphs. The definition of edge rewiring is given as follows:

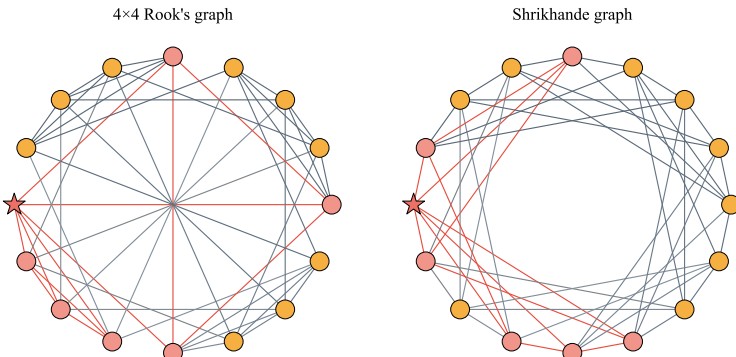

Figure 4: Notable examples of strongly regular graphs: $4 \times 4$ Rook's graph and Shrikhande graph.

**Definition 4.** *(The Edge Rewiring)* *Let $G_1 = (V_1, E_1)$ be a simple undirected graph, we can construct $G_2 = (V_2, E_2)$ by selecting two disjoint edges $(u, v), (x, y) \in E_1$ and replacing them with $(u, y)$ and $(x, v)$, i.e., $V_1 = V_2$ and $E_2 = (E_1 \setminus \{(u, v), (x, y)\}) \cup \{(u, y), (x, v)\}$.*

Through edge rewiring, we can obtain two non-isomorphic graphs with identical degree distributions. Note that, the edge rewiring may alter the eigenvalue distribution of graphs, potentially leading to differences in their von Neumann entropies. When we repeat edge rewiring perturbations on a $d$-regular graph, its distribution can be approximated as a uniform random $d$-regular graph (Kim & Vu, 2003), and the von Neumann entropy ultimately converges to a stable value. Thus, before presenting the proof of Theorem 1, we first state Theorem 2.

**Theorem 2.** *For this specific class of $d$-regular graph with $|V|$ nodes, the number of edge rewirings required for the von Neumann entropy to converge is upper bounded by $\mathcal{O}(|V| \log |V|)$.*

*proof.* For a fixed degree $d$, performing approximately $O(|V| \log |V|)$ perturbations on a $d$-regular graph is sufficient to approximate the uniform distribution over all $d$-regular graphs (Kim & Vu, 2003). We assume that these perturbations are implemented through the edge rewiring, then the distribution of the resulting regular graphs gradually converges to that of uniformly random $d$-regular graphs after $O(|V| \log |V|)$ times of edge rewiring operations.

Then, let $G$ be a random $d$-regular graph. As the number of nodes $n \to \infty$, the spectral density $\rho_d(\lambda)$ of its adjacency matrix converges to the Kesten-McKay distribution (McKay, 1981):

$$\rho_d(\lambda) = \begin{cases} \dfrac{d}{2\pi} \cdot \dfrac{\sqrt{4(d-1) - \lambda^2}}{d^2 - \lambda^2}, & |\lambda| \le 2\sqrt{d-1} \\ 0, & \text{otherwise} \end{cases} \quad (21)$$

The von Neumann entropy can be expressed as $\mathcal{H}_{VN} = -\int \rho_d(\lambda) \cdot \lambda \log \lambda \, d\lambda$. Therefore, as the spectral density converges, the von Neumann entropy also approaches to a steady value. In other words, once the spectral entropy stabilizes, further edge rewiring operations have no significant impact on its value. Hence, Theorem 2 holds.

Building on our earlier discussion of the ConD-MPNN, its enhanced distinguishing power over the 3-WL test stems from the role of the von Neumann entropy of node contexts. When a graph pair contains the distinct von Neumann entropy contexts, the ConD-MPNN is able to distinguish between them. Consequently, as directly derived from Theorem 2, the number of regular graphs distinguishable by the ConD-MPNN is approximately equal to the number of distinct regular graphs that can be generated from different contextual von Neumann entropies. For Theorem 1, suppose that we start from an initial $d$-regular graph $G_1$, and perform $T$ times of edge rewiring operations to obtain a sequence of graphs $G_1, G_2, \ldots, G_{T+1}$, where each $G_i$ is non-isomorphic to all previous graphs. The total number of pairwise non-isomorphic graph pairs is given by:

$$\binom{T+1}{2} = \frac{(T+1)T}{2}. \quad (22)$$

When the times of edge rewiring is $O(|V| \log |V|)$, then the number of non-isomorphic graph pairs grows as $O(|V|^2 \log^2 |V|)$. Thus, Theorem 1 holds.

## A.5 THE VARIANTS OF K-HOP MPNN: COND-KGIN AND KERA-KGIN

Our proposed framework can be seamlessly integrated into the K-hop MPNN. Notably, the K-hop contexts are naturally obtained during the extraction of K-hop neighboring nodes, eliminating the need for additional preprocessing steps. We first introduce the ConD-KMPNN, a variant of the ConD-based framework built upon the K-hop MPNN architecture. The ConD-KMPNN incorporates the $k$-hop node contexts into the message passing process of a K-hop MPNN. At the $t + 1$-th layer, the node representation for the $k$-hop neighboring aggregation is updated as follows:

$$
\begin{aligned}
\boldsymbol{h}_u^{t+1,k} &= \mathbf{U}^{t,k}(\boldsymbol{h}_u^t, \boldsymbol{m}_u^{t+1,k}), \\
\boldsymbol{m}_u^{t+1,k} &= \sum\nolimits_{v \in \mathcal{N}_u^k} \mathbf{M}^{t,k}(\boldsymbol{h}_u^t, \boldsymbol{h}_v^t, \phi(\mathcal{H}_{VN}(S_u^k)), \phi(\mathcal{H}_{VN}(S_v^k)), \boldsymbol{e}_{uv}^k).
\end{aligned}
\tag{23}
$$

Furthermore, the KerA-based framework can also be integrated into the K-hop MPNN. In this variant, termed KerA-KMPNN, the kernel values are computed between the $k$-hop node contexts of the central node and those of its neighbors. At the $t + 1$-th iteration, the node representation can be updated based on the aggregation of $k$-hop neighboring information as follows:

$$
\boldsymbol{h}_u^{t+1,k} = \mathbf{U}^{t,k}(\boldsymbol{h}_u^t, \boldsymbol{m}_u^{t+1,k}), \quad \boldsymbol{m}_u^{t+1,k} = \sum\nolimits_{v \in \mathcal{N}_u^k} \mathbf{M}^{t,k}(\boldsymbol{h}_u^t, \boldsymbol{h}_v^t, \mathcal{K}(S_u^k, S_v^k), \boldsymbol{e}_{uv}^k),
\tag{24}
$$

where $\mathcal{K}(S_u^k, S_v^k)$ denotes the reproduce kernel between $S_u^k$ and $S_v^k$ discussed in Section 5.

## A.6 PROOF OF THE PROPERTY OF REPRODUCING KERNEL

Intuitively, our kernel is a bounded, symmetric function that is monotonically decreasing in the entropy distance between $x$ and $y$. In our setting, $x$ and $y$ are entropy-based representations of the structural distributions of the contexts $S_u^k$ and $S_v^k$. Thus, when two node contexts have identical structural distributions, their entropy representations are the same and then $\mathcal{K}(x, y) = 1$. As the structural distributions diverge, the entropy distance $\|x - y\|$ increases and the kernel value gradually drops toward 0. Thus, similar structural distributions produce large kernel values, while dissimilar ones produce small kernel values, which is precisely the notion of structural similarity that we aim to capture.

From a more rigorous mathematical standpoint, the kernel can be represented as the inner product of two feature maps, thereby making its role in capturing similarity explicit. A detailed proof is given below. Based on the definition of the kernel method (Hofmann et al., 2008), the kernel serves as a measure of similarity

$$
\mathcal{K}(x, y) = \langle \Phi(x), \Phi(y) \rangle
\tag{25}
$$

where $\Phi(x)$ is the feature map and maps into the dot product space. For the reproducing kernel $\mathcal{K}(x, y) = e^{-|x-y|}$, we can also represent it as an inner product between feature mappings, which explicitly shows how the kernel quantifies similarity.

**Theorem 3.** *For the reproducing kernel $\mathcal{K}(x, y) = e^{-|x-y|}$, there exists an explicit feature mapping $\phi$ defined by*

$$
\Phi(x) = \left\{ \sqrt{\frac{2}{\pi(1 + \omega^2)}} \cos(\omega x), \ \sqrt{\frac{2}{\pi(1 + \omega^2)}} \sin(\omega x) \right\}_{\omega \geq 0},
\tag{26}
$$

*such that the kernel function can be expressed as the inner product in this feature space.*

*proof.* Since $e^{-|x|}$ is an even function, its Fourier transform is real:

$$
\mathcal{F}[e^{-|x|}](\omega) = \frac{2}{1 + \omega^2}.
\tag{27}
$$

Therefore, we can construct a real-valued feature mapping through Fourier transform as

$$
e^{-|x-y|} = \frac{1}{2\pi} \int_{-\infty}^{\infty} \frac{2}{1 + \omega^2} e^{i\omega(x-y)} d\omega = \frac{1}{\pi} \int_0^{\infty} \frac{2 \cos(\omega(x - y))}{1 + \omega^2} d\omega
\tag{28}
$$

Finally, we verify the validity by computing the inner product integral:

$$\langle \Phi(x), \Phi(y) \rangle = \int_0^\infty \left[ \frac{2\cos(\omega x)\cos(\omega y)}{\pi(1+\omega^2)} + \frac{2\sin(\omega x)\sin(\omega y)}{\pi(1+\omega^2)} \right] d\omega$$
$$= \frac{2}{\pi} \int_0^\infty \frac{\cos(\omega x)\cos(\omega y) + \sin(\omega x)\sin(\omega y)}{1+\omega^2} d\omega$$
$$= \frac{1}{\pi} \int_0^\infty \frac{2\cos(\omega(x-y))}{1+\omega^2} d\omega$$
$$= e^{-|x-y|}$$

Thus, the Theorem 3 holds.

### A.7 Proof of Proposition 2

Assume there exists a triangular substructure where nodes $u$ and $v$ are both connected to node $o$. The context entropies of these nodes are $r_u = H_{VN}(S_u^k)$, $r_v = H_{VN}(S_v^k)$, $r_o = H_{VN}(S_o^k)$ respectively. In ConD-MPNN, when entropy values are used as representations, the differences in entropy between the node pairs are defined as

$$\Delta_1 = |H_{VN}(S_u^k) - H_{VN}(S_v^k)|, \Delta_2 = |H_{VN}(S_u^k) - H_{VN}(S_o^k)|, \Delta_3 = |H_{VN}(S_v^k) - H_{VN}(S_o^k)|. \tag{29}$$

In KerA-MPNN, the kernel-based representation for node $u$ is given by

$$\phi\{\mathcal{K}(S_u^k, S_v^k), \mathcal{K}(S_u^k, S_o^k)\}, \tag{30}$$

where we assume the function $\phi$ is *sum*. The difference in kernel-based representation between node $u$ and $v$ is

$$\sigma_2 = |z_u - z_v|$$
$$= |(\mathcal{K}(S_u^k, S_v^k) + \mathcal{K}(S_u^k, S_o^k)) - (\mathcal{K}(S_v^k, S_u^k) + \mathcal{K}(S_v^k, S_o^k))|$$
$$= |\mathcal{K}(S_u^k, S_o^k) - \mathcal{K}(S_v^k, S_o^k)| \tag{31}$$
$$= |\frac{1}{2}e^{-\Delta_2} - \frac{1}{2}e^{-\Delta_3}|.$$

Next, assuming that the function $f(x) = e^{-x}$, and applying the Mean Value Theorem, we obtain:

$$e^{-\Delta_2} - e^{-\Delta_3} = -e^{-c}(\Delta_2 - \Delta_3), \tag{32}$$

where $c \in [\Delta_2, \Delta_3]$, i.e., $c > 0$. Consequently, we have:

$$\frac{1}{2}|e^{-\Delta_2} - e^{-\Delta_3}| = |\frac{1}{2}e^{-c}(\Delta_2 - \Delta_3)| < |\Delta_2 - \Delta_3|. \tag{33}$$

Furthermore, by applying the absolute value inequality, we have the following inequality as

$$|\Delta_2 - \Delta_3| = ||H_{VN}(S_u^k) - H_{VN}(S_o^k)| - |H_{VN}(S_v^k) - H_{VN}(S_o^k)||$$
$$\leq |H_{VN}(S_u^k) - H_{VN}(S_v^k)| = \sigma_1. \tag{34}$$

Then, we can conclude that

$$|(\mathcal{K}(S_u^k, S_v^k) + \mathcal{K}(S_u^k, S_o^k)) - (\mathcal{K}(S_v^k, S_u^k) + \mathcal{K}(S_v^k, S_o^k))| < |H_{VN}(S_u^k) - H_{VN}(S_v^k)|. \tag{35}$$

Therefore, $\sigma_2 < \sigma_1$, the Proposition 2 holds.

### A.8 The time, space complexity comparison

In this section, we compare the time and space complexity of our models with several representative baselines, including the standard MPNN (Xu et al., 2019), Subgraph MPNN (Zhang & Li, 2021), I²-GNN (Huang et al., 2023), PPGN (Maron et al., 2019a), and 1-2-3 GNN (Morris et al., 2019). The subgraph MPNN introduces an additional factor of $s$ in complexity due to subgraph-level GNN, where $s$ denotes the maximum subgraph size. The $d$ denotes the average degree. While the other methods enhance expressive power, they do so at the cost of significantly increased time and space complexity. In contrast, our proposed models, ConD-MPNN and KerA-MPNN, maintain linear space and time complexity. Moreover, both theoretical analysis and empirical results demonstrate that our models achieve expressive power comparable to that of these more complex architectures. This highlights the scalability of our framework while preserving strong expressive power.

Table 7: Comparison of time and space complexity among different models.

| Model | MPNN | Subgraph MPNN | I$^2$-GNN | PPGN | 1-2-3 GNN | ConD-MPNN | KerA-MPNN |
|---|---|---|---|---|---|---|---|
| Space complexity | $\mathcal{O}(|V|)$ | $\mathcal{O}(s|V|)$ | $\mathcal{O}(s|V| \cdot d)$ | $\mathcal{O}(|V|^2)$ | $\mathcal{O}(|V| \cdot d^2)$ | $\mathcal{O}(|V|)$ | $\mathcal{O}(|V|)$ |
| Time complexity | $\mathcal{O}(|V| \cdot d)$ | $\mathcal{O}(s|V| \cdot d)$ | $\mathcal{O}(s|V| \cdot d^2)$ | $\mathcal{O}(|V|^3)$ | $\mathcal{O}(|V| \cdot d^3)$ | $\mathcal{O}(|V| \cdot d)$ | $\mathcal{O}(|V| \cdot d)$ |

## A.9 ADDITIONAL EXPERIMENTAL INFORMATION

In this section, we provide further information about the datasets, describe the implementation details, and report additional experimental results to support our main findings.

### A.9.1 DATASETS

Table 8 summarizes the key statistics of the graph datasets used in our experiments. For the graph classification task, we conduct extensive evaluations on seven standard graph datasets. The datasets are extracted from Small Molecules(Mole), Bioinformatics(Bio), and Social Networks(SN) (Morris et al., 2020a). For evaluation of the expressive power, we use three synthetic benchmarks (e.g., EXP (Abboud et al., 2021), SR25 (Balcilar et al., 2021), CSL (Murphy et al., 2019)) and BREC (Wang & Zhang, 2024) datasets that are indistinguishable by the 1-4 WL tests. For the graph regression task, we evaluate on the QM9 (Ramakrishnan et al., 2014) and ZINC (Dwivedi et al., 2023) datasets. We also use the Peptides dataset (Dwivedi et al., 2022) to evaluate the scalability. These datasets exhibit diverse characteristics in terms of the number of graphs, nodes, edges, and classes, thereby enabling a comprehensive evaluation of model scalability, expressive power, and generalization across various domains and graph structures.

Table 8: Information of the graph datasets used in our experiments.

| Datasets | # of graphs | Mean # nodes | Mean # edges | Classes | Description |
|---|---|---|---|---|---|
| MUTAG | 188 | 17.93 | 19.79 | 2 | Mole |
| PROTEINS | 1113 | 39.06 | 72.82 | 2 | Bio |
| PTC_MR | 344 | 14.29 | 14.69 | 2 | Mole |
| COLLAB | 5000 | 74.49 | 2457.78 | 3 | SN |
| IMDB-B | 1000 | 19.77 | 96.53 | 2 | SN |
| IMDB-M | 1500 | 13.00 | 65.94 | 3 | SN |
| DD | 1178 | 284.32 | 715.66 | 2 | Bio |
| Peptides-Func | 15,535 | 150.94 | 307.30 | 10 | Mole |
| EXP | 1000 | 5.00 | 5.00 | 10 | Synthetic |
| SR25 | 500 | 25.00 | 50.00 | 2 | Synthetic |
| CSL | 150 | 10.00 | 20.00 | 10 | Synthetic |
| QM9 | 130831 | 18.00 | 19.00 | 12 | Mole |
| ZINC | 12000 | 23.15 | 24.99 | 1 | Mole |

### A.9.2 EXPERIMENTAL SETTING

This section mainly introduces the implementation details for each model.

For **ConD-WL**, we follow the same experimental setup as the original WL test (Shervashidze et al., 2011). We conduct the 10-fold cross-validation, selecting one fold from the training set for validation in each run. The experiments are repeated ten times, and the average performance along with the standard deviation is reported. For TU datasets, we search the number of hop $k$ in $\{1, 2, 3, 4\}$, the hyperparameter $C$ of the C-SVM classifier in $\{1e-7, 1e-5, 1e-3, 1e-1, 1e+1, 1e+3, 1e+5, 1e+7\}$.

For **ConD-GIN** and **KerA-GIN**, we follow the same experimental setup as the original GIN (Xu et al., 2019). Specifically, we conduct 10-fold cross-validation and report the average test accuracy across all folds. For each model, we select the epoch with the highest mean accuracy and report its corresponding standard deviation. For TU datasets, we follow the exact hyperparameter search range employed in the original GIN implementation to ensure a fair comparison. For the EXP, CSL and SR datasets, we search the $k$ in $\{1, 2, 3, 4\}$. We set the sampling number for KerA-GIN to the minimum number of nodes among all graphs in the dataset. When the number of neighbor nodes

is insufficient, we apply zero-padding to maintain consistent input dimensions. Specifically, for the CSL and EXP datasets, we repeat each experiment four times and report the average results. For the SR dataset, we run the experiment once and report the best result, following the same protocol used in prior work. For all datasets, the dimensionality of the linear transformation $\phi$ matches the feature dimension of $h$ in the hidden layer, and is selected from the set $\{16, 32, 64, 128\}$ through a search process.

For **ConD-KGIN** and **KerA-KGIN**, we follow the same experimental setup as KP-GIN (Feng et al., 2022) to ensure a fair comparison. Specifically, we adopt the exact hyperparameter search ranges used in KP-GIN. For the TU datasets and the three synthetic benchmarks, the implementation details are identical to those used in the corresponding GIN-based variants. For the ZINC dataset, we search the node context size in $\{8, 16\}$, the number of layers in $\{9, 17\}$, and the hidden feature dimension in $\{94, 106\}$. For the QM9 dataset, the hidden dimension is 128 and we search the number of layers in $\{8, 16\}$. For the Peptides dataset, we search the node context size in $\{4, 6, 8, 10\}$, the number of layers in $\{4, 6, 8, 10\}$. For all datasets, the dimensionality of the linear transformation $\phi$ also matches the hidden dimension. We evaluate all methods using two context construction strategies, i.e., the shortest path distance and random walk. For the BREC dataset, we set the context size is 4.

### A.9.3 COMPARISON WITH DIFFERENT BACKBONE

To more comprehensively evaluate the effectiveness of our proposed framework, we further instantiate both methodologies within the K-GCN and K-GraphSAGE backbones, as reported in Table 9. We compare classification performance against the KP-GCN and KP-GraphSAGE baselines Feng et al. (2022) across five datasets. Overall, ConD-KGCN and KerA-KGCN achieve comparable or slightly better performance than KP-GCN on most datasets; for example, KerA-KGCN attains the highest mean accuracy on DD, PROTEINS, and IMDBBINARY. Similarly, ConD-KGraphSAGE and KerA-KGraphSAGE match or slightly exceed the performance of KP-GraphSAGE Feng et al. (2022), with KerA-KGraphSAGE achieving the best accuracy on four out of the five datasets. These results indicate that the proposed context distribution can be seamlessly integrated into different message passing backbones while maintaining competitive performance, and they further suggest that kernel-based aggregation is effective in mitigating distributional discrepancies between contexts rooted at neighboring nodes.

Table 9: Comparison under different backbones.

| Method | MUTAG | DD | PTC-MR | PROTEINS | IMDBBINARY |
|---|---|---|---|---|---|
| KP-GCN | 91.7±6.0 | 79.0±4.7 | **67.1±6.3** | 75.8±3.5 | 75.9±3.8 |
| **ConD-KGCN** | 92.2±6.5 | 79.9±4.2 | 65.9±8.9 | **76.5±3.7** | 75.9±4.4 |
| **KerA-KGCN** | 92.8±5.3 | 80.3±3.5 | 66.8±6.2 | **76.6±4.7** | **76.0±4.3** |
| KP-GraphSAGE | 91.7±6.5 | 78.1±2.6 | 66.5±4.0 | **76.5±4.6** | 76.4±2.7 |
| **ConD-KGraphSAGE** | 92.2±5.5 | 77.6±3.2 | 67.4±7.9 | 76.3±5.2 | 76.1±5.9 |
| **KerA-KGraphSAGE** | 92.8±5.3 | 78.7±3.6 | 67.1±6.6 | 75.9±4.5 | **76.5±3.0** |

### A.9.4 SCALABILITY ON LARGE-SCALE GRAPHS

Table 10: Performance on the Peptides dataset.

| Method | Peptides (Test AP ↑) |
|---|---|
| GIN | $0.6555 \pm 0.0088$ |
| K-GIN | $0.6479 \pm 0.0041$ |
| GIN + CSE | $0.6619 \pm 0.0077$ |
| HyMN | $0.6857 \pm 0.0055$ |
| GT+MoSE | $0.6350 \pm 0.0110$ |
| **ConD-KGIN(ours)** | $\mathbf{0.6901 \pm 0.0034}$ |

To evaluate the scalability on large-scale graphs, we compare the performance of different methods on the Peptides dataset. The results in Table 10 show that the ConD-KGIN method achieves the highest score, with an AP score of $0.6901 \pm 0.0034$. The ConD-KGIN method outperforms all other models, including the GIN (Xu et al., 2019), the K-GIN(Nikolentzos et al., 2020), the GIN +

CSE (Southern et al., 2025), the HyMN (Southern et al., 2025) and GT+MoSE (Bao et al., 2025), demonstrating its superior performance in this setting.

### A.9.5 VISUALIZATION OF ADDITIONAL NODES

To further illustrate the effect of our kernel-based representation, we visualize the context distributions of selected nodes from the same graph in the DD dataset. Specifically, we sample nodes with IDs $\{0, 5, 10, 15\}$.

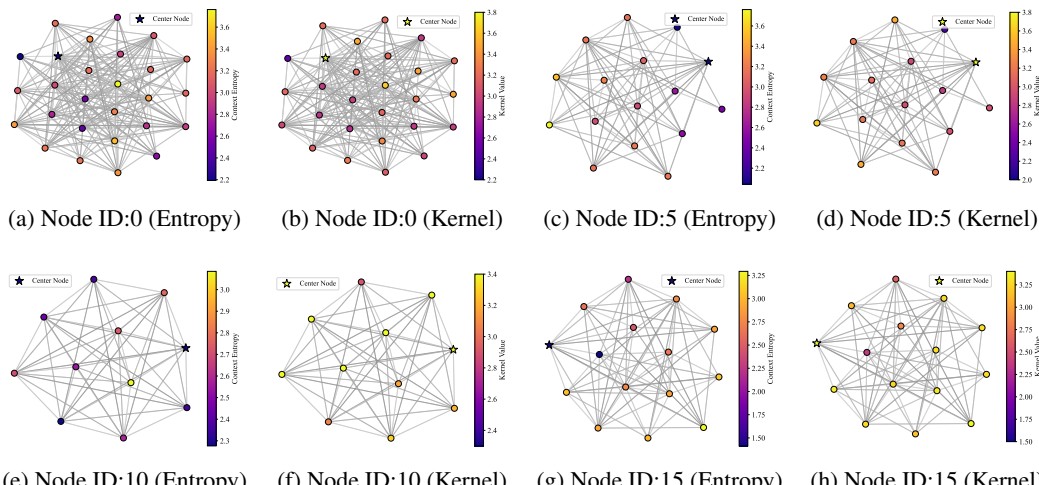

| (a) Node ID:0 (Entropy) | (b) Node ID:0 (Kernel) | (c) Node ID:5 (Entropy) | (d) Node ID:5 (Kernel) |
|---|---|---|---|
| (e) Node ID:10 (Entropy) | (f) Node ID:10 (Kernel) | (g) Node ID:15 (Entropy) | (h) Node ID:15 (Kernel) |

Figure 5: Additional visualizations of entropy-based and kernel-based node context distributions.

As shown in Figure 5, the entropy-based representations exhibit the scale variation across these nodes, with values spanning a wide range. In contrast, the kernel-based representations are more concentrated, demonstrating reduced inter-node discrepancy. This observation consistently supports our claim that the reproducing kernel effectively smooths and normalizes structural context, thereby facilitating more stable message passing in GNNs.

Table 11: Accuracy (%) on synthetic datasets with varying context size $K$.

| Method | EXP (ACC) | | | | SR (ACC) | | | | CSL (ACC) | | | |
|---|---|---|---|---|---|---|---|---|---|---|---|---|
| | K=1 | K=2 | K=3 | K=4 | K=1 | K=2 | K=3 | K=4 | K=1 | K=2 | K=3 | K=4 |
| K-GIN | 50 | 50 | 100 | 100 | 6.67 | 6.67 | 6.67 | 6.67 | 12 | 22.7 | 42 | 62.7 |
| **ConD-GIN** | 50 | 100 | 100 | 100 | 100 | 100 | 100 | 100 | 22 | 52 | 90 | 100 |
| **KerA-GIN** | 50 | 100 | 100 | 100 | 100 | 100 | 100 | 100 | 22 | 52 | 90 | 100 |

### A.9.6 IMPACT OF CONTEXT SIZE ON EXPRESSIVE POWER AND GRAPH CLASSIFICATION

Table 12: Performance comparison with varying context size $k$.

| Metric | Method | PROTEINS | | | DD | | |
|---|---|---|---|---|---|---|---|
| | | $k=2$ | $k=3$ | $k=4$ | $k=2$ | $k=3$ | $k=4$ |
| Accuracy | GNN-AK+ | 76.5±3.4 | 76.0±3.9 | 75.5±4.8 | OOM | OOM | OOM |
| | KP-GIN | 75.9±4.2 | 75.3±4.7 | 75.8±5.3 | 78.6±2.9 | 78.0±4.4 | **80.0±5.3** |
| | **ConD-KGIN** | **76.8±3.4** | **76.6±5.2** | **77.0±3.1** | 79.7±3.0 | 79.5±3.5 | 78.9±3.9 |
| | **KerA-KGIN** | 76.7±6.5 | 75.8±4.7 | 75.9±4.9 | **80.4±4.9** | **79.7±5.2** | 79.6±5.6 |
| Training Time (s/epoch) | GNN-AK+ | 0.789 | 0.791 | 0.786 | OOM | OOM | OOM |
| | KP-GIN | 0.461 | 0.646 | 0.589 | **0.944** | **1.037** | **1.283** |
| | **ConD-KGIN** | **0.421** | **0.468** | 0.501 | 1.176 | 1.237 | 1.561 |
| | **KerA-KGIN** | 0.447 | 0.482 | **0.494** | 1.094 | 1.312 | 1.358 |

We also conduct an analysis to investigate how varying the context size $k$ influences the expressive power of the models. Specifically, we vary $k$ from $\{1, 2, 3, 4\}$ and evaluate the classification accuracy using the K-GIN backbone. The results are presented in Table 11. It can be observed that

our proposed models consistently outperform the original backbone across all values of $k$. As $k$ increases, the expressive power of the models improves accordingly. When $k = 4$, both ConD-GIN and KerA-GIN achieve perfect accuracy on all three synthetic datasets. Specifically, even when $k = 1$, our models can distinguish all non-isomorphic graphs in the SR dataset. This further demonstrates the effectiveness of context von Neumann entropy as a powerful structural representation.

Furthermore, we conduct experiments on the PROTEINS and DD datasets by varying the context size from 2 to 4 as shown in Table 12, both ConD-KGIN and KerA-KGIN maintain strong performance across different $k$.

Table 13: Accuracy (%) with different context construction strategies.

| Method | EXP | | SR | | CSL | |
|---|---|---|---|---|---|---|
| | SPD | RW | SPD | RW | SPD | RW |
| ConA-KGIN | $100 \pm 0.0$ | $100 \pm 0.0$ | $100 \pm 0.0$ | $100 \pm 0.0$ | $98.67 \pm 4.2$ | $100 \pm 0.0$ |
| KerA-KGIN | $100 \pm 0.0$ | $100 \pm 0.0$ | $100 \pm 0.0$ | $100 \pm 0.0$ | $92.00 \pm 2.8$ | $100 \pm 0.0$ |

### A.9.7 ANALYSIS ON CONTEXT CONSTRUCTION STRATEGY

In this experiment, we compare the expressive power of two methods, the ConA-KGIN and the KerA-KGIN, using various context construction strategies, including the shortest-path distance (SPD) and the random walk (RW). As shown in Table 13, both methods achieve 100% performance on the EXP and SR datasets across all context types. Additionally, using the random walk strategy to construct the context also achieves 100% accuracy on the CSL dataset.

Table 14: Comparison of KerA-KGIN and KerA-KGIN without sampling on benchmark datasets.

| Metric | Method | MUTAG | PTC | PROTEINS | IMDBBINARY | DD |
|---|---|---|---|---|---|---|
| Test Accuracy | KerA-KGIN | $\mathbf{92.8 \pm 7.0}$ | $\mathbf{67.6 \pm 6.4}$ | $\mathbf{76.7 \pm 6.5}$ | $\mathbf{76.6 \pm 4.1}$ | $\mathbf{80.4 \pm 4.9}$ |
| | KerA-KGIN w/o sample | $92.2 \pm 5.4$ | $65.9 \pm 8.0$ | $76.3 \pm 5.3$ | $76.2 \pm 4.0$ | $79.0 \pm 3.5$ |
| Time (s) | KerA-KGIN | $\mathbf{0.179}$ | $\mathbf{0.265}$ | $\mathbf{1.014}$ | $\mathbf{0.889}$ | $\mathbf{4.919}$ |
| | KerA-KGIN w/o sample | $0.185$ | $0.325$ | $1.170$ | $1.015$ | $126.980$ |

### A.9.8 ANALYSIS ON SAMPLING STRATEGY

As our KerA-based method employs an efficient neighbor sampling mechanism, we perform a detailed ablation study on this component, as presented in Table 14. We evaluate the test accuracy and the inference time respectively. When the sampling strategy is removed from KerA-KGIN, the accuracy is lower across most datasets. More importantly, our KerA-based method with neighbor sampling achieves a substantial improvement in computational efficiency. For instance, on the DD dataset, the training time is reduced from 126.980s to 4.919s. These results confirm the effectiveness of the proposed neighbor sampling mechanism, which not only preserves the expressive power of the model but also significantly boosts runtime efficiency—particularly on large-scale graphs.

To validate the effectiveness of the proposed sampling strategy, we test different sampling strategies on graph classification datasets. Specifically, we sample high-entropy nodes, low-entropy nodes, and performed random sampling. As shown in the experimental results in Table 15, the high-entropy nodes sampling strategy provides the best performance on both datasets, particularly on the DD dataset. Thus, nodes with higher entropy have more diverse connections, offering richer structural information. By prioritizing these neighbors, the model captures the most informative parts of the graph, enhancing message passing.

Table 15: Performance of different sampling strategies.

| Dataset | High-entropy | Low-entropy | Random Sampling |
|---|---|---|---|
| IMDB-BINARY | $\mathbf{76.6 \pm 4.1}$ | $75.4 \pm 3.3$ | $75.1 \pm 5.0$ |
| DD | $\mathbf{80.4 \pm 4.9}$ | $78.7 \pm 5.3$ | $77.3 \pm 6.9$ |

Furthermore, we conduct the sensitivity analysis experiments on two representative benchmark datasets to evaluate the impact of varying the number of sampled nodes. The results on the DD and IMDB-BINARY datasets are shown in Table 16. We observe that the performance remains relatively stable across different values of $m$, indicating that our method is not overly sensitive to the specific choice of sampling size. In addition, Table 16 also shows that when the neighboring sampling number $m = 4$, our model achieves the best performance in the Table 3. This indicates that a small sampling number is sufficient to capture the essential structural information, and larger number do not necessarily lead to further improvements.

Table 16: Accuracy (%) with different sampling number $m$.

| Dataset | $m$=2 | $m$=4 | $m$=6 | $m$=8 |
|---|---|---|---|---|
| DD | 79.3±5.6 | 79.6±3.3 | 79.5±3.3 | 78.6±5.4 |
| IMDB-BINARY | 75.3±2.9 | 76.8±2.9 | 75.9±3.3 | 76.4±3.8 |

### A.9.9 COMPARISONS WITH DIFFERENT ENTROPY CHOICES

As we discussed in A.3, we provided a theoretical justification for the use of von Neumann entropy. In this section, we conduct empirical experiments to further verify the advantages of incorporating von Neumann entropy as a structural representation. Table 17 shows that, within our framework, the variant using von Neumann entropy is more expressive than the variant using structural Shannon entropy. Table 18 presents the corresponding comparison on graph classification tasks between structural Shannon entropy (see Definition 3) and von Neumann entropy. The results are showing that both von Neumann entropy and structural Shannon entropy are effective components within our framework and achieve comparable performance on graph classification tasks, with von Neumann entropy exhibiting a slight advantage in terms of mean accuracy and smaller variance.

Table 17: Comparison of different entropy on expressive power.

| Method | EXP | CSL | SR |
|---|---|---|---|
| Structural Shannon Entropy | 100.0 | 26.67 | 96.7 |
| **von Neumann entropy** | **100.0** | **100.0** | **100.0** |

### A.9.10 COMPARISONS WITH DIFFERENT KERNEL FUNCTIONS

In our proposed KerA-based framework, we adopt a reproducing kernel for the kernel aggregation module. To validate its effectiveness, we compare it against the context Jensen-Shannon kernel (Bai & Hancock, 2013), a widely used baseline for measuring distributional similarity. The comparison results are shown in Table 18. We can observe that the reproducing kernel consistently outperforms the Jensen-Shannon kernel across all five graph classification datasets. This result indicates that the reproducing kernel more effectively alleviates the discrepancy between context entropies rooted at neighboring nodes, leading to more stable and informative message aggregation, which is consistent with our theoretical analysis.

Table 18: Comparison of different entropy and kernel strategies on five datasets.

| Category | Method | MUTAG | D&D | PTC-MR | PROTEINS | IMDBBINARY |
|---|---|---|---|---|---|---|
| *Entropy* | Structural Shannon Entropy | 92.8±4.6 | 79.0±3.5 | 66.8±9.4 | 76.4±3.4 | 75.5±3.3 |
| | **von Neumann entropy** | **93.9±5.5** | **79.7±3.0** | **67.4±9.4** | **77.0±3.1** | **76.3±4.3** |
| *Kernel* | Jensen Shannon kernel | 92.8±7.0 | 79.5±5.8 | 65.6±6.9 | 76.2±4.0 | 75.9±4.3 |
| | **Reproduce kernel** | **92.8±4.6** | **80.4±4.9** | **65.9±8.0** | **76.7±6.5** | **76.6±4.1** |

### A.10 THE USE OF LARGE LANGUAGE MODELS (LLMS)

In this work, Large Language Models (LLMs) are only employed to assist with language polishing and writing refinement. The LLM did not influence content ideation, data analysis, or experimental design in any way.

