# OpenReview forum: "From Contextual Distributions to Messages: Entropy-Guided GNNs"
_ICLR.cc/2026/Conference — Submitted to ICLR 2026_

### Official Review · Reviewer_AR9t · 2025-10-15

**Soundness:** 3
**Presentation:** 3
**Contribution:** 2
**Rating:** 4
**Confidence:** 4

**Summary:**

This paper proposes ConD-MPNN, a framework that enhances GNN expressiveness by propagating entropy-encoded representations of node contexts instead of performing message passing on k-tuples or subgraphs. The method claims to achieve better expressiveness than 1-WL while maintaining linear complexity. A kernel-based variant is introduced to handle distributional discrepancies. Experiments on synthetic expressiveness benchmarks and real-world datasets show competitive performance.

**Strengths:**

1. The paper clearly identifies the computational bottleneck of subgraph GNNs and higher-order methods, and the proposed entropy-based encoding is conceptually sound as a compact structural representation.

2. The paper includes evaluations on synthetic benchmarks, real-world graph classification, and regression tasks. The ablation studies and scalability experiments across different backbones  demonstrate thoroughness.

3. The visualizations in Figure 3-4 effectively illustrate how the method works.

**Weaknesses:**

1. The paper positions itself as an efficient alternative to subgraph GNNs but fails to compare against recent structural encoding methods that make identical efficiency claims, such as [1-3]. Without these comparisons, it is hard to evaluate whether the proposed method actually provides advantages over state-of-the-art structural encodings.

2. Definition 1's "k-hop context construction using shortest path distance" appears to similar to previous studies such as [1,4], which explicitly constructs contexts at distance k and processes k-hop neighborhoods separately.

3. The claim that "training is $O(|V|·d)$, same as standard MPNN" is misleading because it ignores that features are augmented to dimension $d^k$, making true training complexity $O(|E|d^{k})$. In the experiments, it would be better to provide empirical comparison with the structural encoding based methods, as they also aim to improve the computational efficiency, which share the similar goal with the proposed method.

4. Claiming "strictly more powerful than 1-WL and 2-WL" is baseline expectation, many 2023-2024 papers exceed this. The field has moved toward quantitative characterizations (which specific cycles/cliques can be counted at which levels) rather than qualitative WL comparisons. This point makes this work incremental.

5. The work also missed some direct baselines, such as the structural encoding based methods mentioned earlier and 2-FGNN [5], to name a few.

---

**Reference**

1. Yao et al., Improving the Expressiveness of 𝐾-hop Message-Passing GNNs by Injecting Contextualized Substructure Information

2. Bao et al., Homomorphism Counts as Structural Encodings for Graph Learning

3. Kanatsoulis et al., Learning Efficient Positional Encodings with Graph Neural Networks

4. Abbound et al., Shortest Path Networks for Graph Property Prediction

5. Zhang et al., Beyond Weisfeiler-Lehman: A Quantitative Framework for GNN Expressiveness

**Questions:**

1. Can you provide direct empirical comparisons against structural encoding based methods in terms of efficiency and performance.

2. The $O(|V|·d^{3k})$ preprocessing, for k=3 with typical dimensions, is this prohibitively expensive?

---

> ### Author Response · Authors · 2025-11-24
> **Response to Reviewer AR9t (1)**
>
> Thank you for your valuable comments! In the following section, we address the weaknesses (W) and questions (Q) mentioned above. **The changes in the revised version are marked in blue.**
>
> > W1 &Q1:  The paper positions itself as an efficient alternative to subgraph GNNs but fails to compare against recent structural encoding methods that make identical efficiency claims, such as [1-3]. Without these comparisons, it is hard to evaluate whether the proposed method actually provides advantages over state-of-the-art structural encodings.
>
> We thank the reviewer for providing the latest structural encoding methods. In the revised version, we have added direct comparisons with SEK-GIN on QM9 and GT+MoSE on Peptides-func, and we have reproduced PEARL on BREC. Overall, our methods achieve **competitive or superior performance** compared to these structural encodings.
>
> On the QM9 dataset, we compare with SEK-GIN as：
>
> | Target  | SEK-GIN [1] | ConD-KGIN |
> |----------|----------|-----------|
> | μ     | **0.358**   | 0.360   |
> | α     | 0.228   | **0.224**   |
> | ε_HOMO  | **0.00106** | 0.00237  |
> | ε_LUMO  | 0.00229  | **0.00228**  |
> | Δε    | 0.00335  | **0.00330**  |
> | ⟨R²⟩   | 16.91   | **16.26**   |
> | ZPVE   | **0.00013**  | 0.00018  |
> | U₀    | **0.0587**  | 0.0667   |
> | U     | 0.0672  | **0.0323**   |
> | H     | 0.073   | **0.0344**   |
> | G     | 0.0592  | **0.0309**   |
> | C_v    | 0.0924  | **0.0916** |
>
> These results show that our framework achieves **comparable or better performance** than SEK-GIN on many targets, e.g., α, ε_LUMO, Δε, ⟨R²⟩, U, H, G, C_v)
>
> On the Peptides-func dataset, we compare with GT+MoSE as:
>
> | Model        | Peptides-func (AP ↑) |
> |---------------------|----------------------|
> | GT+MoSE [2]     | 0.635 ± 0.011     |
> | ConD-KGIN (ours)   | **0.690 ± 0.003**   |
>
> Here, ConD-KGIN clearly **outperforms** GT+MoSE.
>
> On the BREC dataset, we reproduce the PEARL and compare with our method as:
>
> | Method               | Basic (60) | Regular (140) | Extension graph (100) | CFI (100) | Total (400) | Accuracy |
> | -------------------- | ---------- | ------------- | --------------------- | --------- | ----------- | -------- |
> | PEARL [3]            | 58         | 51            | 98                    | 1         | 208         | 52%      |
> | **ConD-KGIN (ours)** | **60**     | **100**       | 98                    | **78**    | **336**     | **84%**  |
>
> Since BREC is designed to evaluate **expressive power**, this result indicates that our method provides a substantial advantage over PEARL in this setting.
>
> In addition, we have **updated the Related Work section** in the main text to discuss these methods more clearly.
>
> “the SEK-GIN[1] injects contextualized substructure information to enhance the expressiveness, the PEARL[3] introduces efficient positional encoding, and the MoSE[2]] proposes motif structural encoding as structural encoding based on counting graph homomorphisms”
>
> > W2: Definition 1's "k-hop context construction using shortest path distance" appears to similar to previous studies such as [1,4], which explicitly constructs contexts at distance k and processes k-hop neighborhoods separately.
>
> We appreciate the reviewer for pointing this out. We agree that the approach for constructing contexts is similar, however, it is not exactly the same. We would like to clarify the differences more clearly. In [1], the context of a node is constructed at a fiexd hop radius $k$. In our framework, for each node $u$, we define its context as the union of its 1-hop to $k$-hop neighbors, which is specifically designed to support the subsequent entropy-based context distribution used in our method. Reference [4] is more aligned with the K-hop message passing framework that we adopt as a backbone, rather than the context construction approach itself. Furthermore, constructing the context is a premise for our work, which facilitates the computation of context entropy, but it is not the core contribution. In the revised version, we have added a discussion of [1, 4] in Lines 209-210.

---

> ### Author Response · Authors · 2025-11-24
> **Response to Reviewer AR9t (2)**
>
> > W3: The complexity claim is misleading due to feature dimensionality blow-up and requests empirical efficiency comparisons with structural encoding baselines.
>
> We thank the reviewer for this helpful comment and apologize for the confusion caused by the original wording.
>
> First, Regarding the complexity discussion, our method does **not** explicitly construct a feature vector of dimension $d^k$. In our implementation, each context $S_u^k$ is summarized into a fixed-dimensional entropy, e.g, one dimension, which is then mapped by a linear layer to the same hidden dimension as the backbone.  Thus, the training time of our proposed ConD-MPNN and KerA-MPNN is the same as that of the standard MPNN. Additionally, within the $k$-hop framework, the training time of ConD-KMPNN is the same to that of K-hop MPNN, both being $O(k|V|d)$, where $k$ can be treated as a constant.
>
> Second,  in terms of efficiency, we selected the COLLAB (sufficiently dense) and DD datasets (sufficiently large) as challenging settings. On these large graphs, we found that existing subgraph-based GNNs and structure encoding (KP-GIN) run out of memory on our hardware, which already reflects the higher efficiency and better scalability of our approach.For KP-GIN on COLLAB, the total time exceeds one day only for the preprocessing stage, which further indicates that this method is not scalable to large datasets in practice. To provide a clearer quantitative contrast, we additionally report results on the medium-scale PROTEINS dataset, where our method still shows advantages in both running time and memory consumption.
>
> | Model     | DD Total time (s) | DD Memory | COLLAB Total time (s) | COLLAB Memory | PROTEINS Total time (s) | PROTEINS Memory |
> | --| ---| --- | --- | - | --- | ---|
> | ESAN    | OOM      | OOM       | OOM    | OOM           | 1645 | 2170M           |
> | GNN-AK+    | OOM    | OOM       | OOM       | OOM           | 1821   | 5468M           |
> | KP-GIN     | 19096      | **1116M** | >>1 day       | --            | 1960           | 476M            |
> | **ConD-KGIN (ours)** | **7137**          | 1544M     | 35420         | **2620M**     | **1317**      | **474M**        |
>
> Furthermore, we compare the evaluation time with PEARL[3] on BREC dataset as
>
> | Method        | Evaluation Time (s) | Memory|Accuracy |
> |---|---|--| --|
> | PEARL       |   17448 | 1028M |   52%     |
> | **ConD-KGIN(ours)**      |  **3940**  | **442M** | **84%**     |
>
> The results demonstrates that, compared with the structural encoding method, our approach offers a more favorable trade-off between computational cost and performance.
>
> > W4: The field has moved toward quantitative characterizations rather than qualitative WL comparisons.
>
> We thank the reviewer for this important perspective. The evaluation of a graph neural network’s expressive power typically encompasses several aspects: approximation ability, the ability to distinguish non-isomorphic graphs, substructure counting ability [6] and the ability to identify the cut vertex and cut edge[7]. In particular, our work targets the aspect of expressiveness concerned with **distinguishing non-isomorphic graphs**. On the BREC benchmark, which is explicitly constructed to probe the limits of WL-hierarcy tests and high-expressivity GNNs. In experiments, our proposed method achieves SOTA performance on the BREC dataset, successfully distinguishing graphs that are beyond the reach of 3-WL and even some cases of 4-WL. The results are shown below:
>
> | Method       | Basic (60) | Regular (140) | Extension graph (100) | CFI (100) | Total (400) | Accuracy |
> |----|--|---|------|---|--|--|
> | I²-GNN | 60     | 100   | **100**    | 21     | 281     | 70.2%   |
> | GNN-AK+  | 60     | 50   | 97      | 15     | 222     | 55.5%   |
> | PPGN    | 60     | 50    | 100     | 23     | 233     | 58.2%   |
> | NGNN   | 59     | 48  | 59      | 0     | 166     | 41.5%   |
> | N²GNN    | 60     | 100      | 100   | 27     | 287     | 71.8%   |
> | DSS-GNN      | 58     | 48       | **100**    | 15     | 221     | 55.2%   |
> | K-GIN  | 60     | 48       | 100     | 27     | 235     | 58.8%   |
> | KP-GIN   | 60     | 106      | 98     | 11     | 275     | 68.8%   |
> | **ConD-KGIN (ours)** | **60**   | **100**    | 98   | **78**   | **336**   | **84%**  |
> | **KerA-KGIN (ours)** | **60**   | **118**    | 81      | **72**   | **331**   | **82.75%** |
>
> We view this as the primary contribution of our work, i.e., a context-distribution–based plug-in that can be integrated into standard MPNN backbones while delivering substantial gains on a rigorous expressiveness benchmark and remaining computationally efficient.
>
> We also fully agree with the reviewer that there are more fine-grained, quantitative characterizations. Our future work will focus on exploring the theoretical relationship between the distribution of contextual eigenvalue and substructure counting ability, as well as further enhancing the model’s ability to identify cut vertex and cut edge.

---

> ### Author Response · Authors · 2025-11-24
> **Response to Reviewer AR9t (3)**
>
> > W5: The work also missed some direct baselines, such as the structural encoding based methods mentioned earlier and 2-FGNN [5], to name a few.
>
> In the revised manuscript, we have added comparisons with the structural encoding-based baselines mentioned earlier, both in the experimental section and in the related work.
>
> Regarding 2-FGNN [5], we agree that it is a highly relevant high-expressivity baseline. 2-FGNN is built on the 2-FWL test, which is known to be equivalent in expressive power to 3-WL [8].  As demonstrated in both our theoretical analysis and experiments (e.g., on the BREC benchmark), our method successfully distinguishes graph pairs that are indistinguishable by 3-WL and even certain cases beyond the expressive power of 4-WL. This indicates that, at least on such challenging instances, our framework goes beyond the expressiveness of standard 2-FWL/3-WL–bounded architectures such as 2-FGNN.
>
> > Q1 & Q2: Can you provide direct empirical comparisons against structural encoding based methods in terms of efficiency? The $O(|V|·d^{3k})$ preprocessing, for k=3 with typical dimensions, is this prohibitively expensive?
>
> We thank the reviewer for pointing this out.
>
> First,  this complexity refers to the **offline preprocessing step** used to compute entropy-based context features. In our implementation, this step is performed once before training. Although the theoretical preprocessing complexity can look high in the worst case, our framework remains practically efficient and scalable on challenging benchmarks. When comparing end-to-end time and memory consumption, our method still demonstrates clear advantages. Following the reviewer’s suggestion, we have added **direct empirical efficiency comparisons** with strong subgraph-based and structural encoding methods as below. The total time contains the preprocessing time.
>
> | Model                | DD Total time (s) | DD Memory | COLLAB Total time (s) | COLLAB Memory | PROTEINS Total time (s) | PROTEINS Memory |
> | -------------------- | ----------------- | --------- | --------------------- | ------------- | ----------------------- | --------------- |
> | ESAN                 | OOM               | OOM       | OOM                   | OOM           | 1645                    | 2170M           |
> | GNN-AK+              | OOM               | OOM       | OOM                   | OOM           | 1821                    | 5468M           |
> | KP-GIN               | 19096             | **1116M** | >>1 day               | --            | 1960                    | 476M            |
> | **ConD-KGIN (ours)** | **7137**          | 1544M     | 35420                 | **2620M**     | **1317**                | **474M**        |
>
> In terms of preprocessing time, KP-GIN requires more than one day on COLLAB due to its **heavy preprocessing**, while our method is substantially more efficient. In terms of overall efficiency, ESAN and GNN-AK+ are not scalable for the large and dense datasets.
>
> Therefore, these results indicate that our framework remains practically efficient and scalable. We believe that simultaneously reducing pretraining time, training time, and memory consumption while enhancing expressive power is a significant challenge in the field. This will also be a key direction for our future research. In the conclusion section of the revised version, we have added a discussion of future work.
>
> [1] Yao et al., Improving the Expressiveness of $K$-hop Message-Passing GNNs by Injecting Contextualized Substructure Information
>
> [2] Bao et al., Homomorphism Counts as Structural Encodings for Graph Learning
>
> [3] Kanatsoulis et al., Learning Efficient Positional Encodings with Graph Neural Networks
>
> [4] Abbound et al., Shortest Path Networks for Graph Property Prediction
>
> [5] Zhang et al., Beyond Weisfeiler-Lehman: A Quantitative Framework for GNN Expressiveness
>
> [6] Zhang B, Fan C, Liu S, et al. The expressive power of graph neural networks: A survey[J]. IEEE Transactions on Knowledge and Data Engineering, 2024.
>
> [7] Zhang B, Luo S, Wang L, et al. Rethinking the Expressive Power of GNNs via Graph Biconnectivity[C]//The Eleventh International Conference on Learning Representations.
>
> [8] Huang, N. T. and Villar, S. A short tutorial on the weisfeilerlehman test and its variants. In ICASSP 2021-2021. IEEE International Conference on Acoustics, Speech and Signal Processing (ICASSP), pp. 8533–8537. IEEE, 2021.

---

### Official Review · Reviewer_rgkQ · 2025-10-16

**Soundness:** 3
**Presentation:** 1
**Contribution:** 2
**Rating:** 2
**Confidence:** 4

**Summary:**

This paper proposes ConD-MPNN, a message-passing framework that augments node features with a summary of each node’s k-hop “context” encoded by the von Neumann entropy. A kernelized variant (KerA-MPNN) aggregates messages using a reproducing kernel defined on these context entropies. The paper claims (i) strictly higher expressive power than 1-WL and 2-WL, with the ability to distinguish “many” regular graphs that 3-WL fails on; (ii) comparable training-time complexity to standard MPNNs (messages depend on precomputed scalars); and (iii) improved performance on several benchmarks.

**Strengths:**

S1: The paper proposes a simple, general plug-in that augments messages with a scalar contextual statistic (entropy), which is architecturally lightweight and could be dropped into many MPNN variants.

S2: The approach achieves many best-in-column results on QM9 targets, and reports very strong BREC performance -- validating its expressiveness claims and demonstrating its in practice usefulness.

**Weaknesses:**

W1: The presentation, motivation and methodology are difficult to follow.
W1a: Figure 1 is not self-contained and does not aid understanding. The caption fails to explain the illustration, what the different node colors denote, or what the histograms represent.
W1b: Figure 2 has the same shortcomings as Figure 1 and likewise adds little to the paper; both read as fillers rather than explanatory figures.
W1c: The theoretical statements are informal. For example, Theorem 2 begins “For this specific class of d-regular graphs with V nodes” but never defines the class. Theorem 3 mentions “significantly different node context distributions” informally and concludes that the method “can effectively reduce these differences,” which is also vague. Theorem 1 should be a proposition: by adding auxiliary information to a 1-WL GNN one can trivially subsume the 1-WL test and, as the 1-WL test equals the 2-WL, gnn can subsume it  as well.
W1d: It is unclear if the example in Figure 3 is representative or an outlier. Instead one would need to plot a measurement over multiple nodes.

W2: Unsupported efficiency claims. The assertion of “significantly reduced computational and memory costs” is not backed by end-to-end time and memory comparisons. The asymptotic discussion in the appendix is likewise unconvincing: since the subgraph size s in subgraph GNNs can be fixed as a constant, their asymptotic computational complexity matches that of the proposed method. The claim is therefore unsupported both asymptotically and empirically.

W3: Reproducibility gaps. The reproducibility statement promises code in the supplementary materials, but no link is provided. Important hyperparameters and data-processing choices are missing or scattered across the appendix. This is especially concerning given the very strong BREC results.

W4: Related work placement. The main paper lacks an in-depth comparison with competitive methods; a related-work section can be is found in the appendix, which reinforces the broader presentation issues.

**Questions:**

Q1: Could you please formalize Theorems 2 and 3 (including precise definitions/metrics)?

Q2: Can you please report total runtime (including preprocessing) and peak memory vs. subgraph GNNs and k-GNN baselines on (a) a large sparse graph and (b) a dense graph, to support the “significant reduction” claim.

Q3: For Figure 3, how was the center node chosen and why 14 neighbors? Is this example representative or an outlier? Could you aggregate this analysis across many nodes/graphs (e.g., distribution of entropy spread before/after kernel mapping)?

Q4: Can you provide an anonymous link for the code so that reviewers can verify the performance (especially BREC)?

---

> ### Author Response · Authors · 2025-11-24
> **Response to Reviewer rgkQ (1)**
>
> Thank you for your valuable comments! In the following section, we address the weaknesses (W) and questions (Q) mentioned above. **The changes in the revised version are marked in blue.**
>
> > W1a: Figure 1 is not self-contained and does not aid understanding. The caption fails to explain the illustration, what the different node colors denote, or what the histograms represent.
>
> Thank you for your valuable feedback. In the original version, different node colors were intended to represent the distinct contexts extracted around different center nodes, and the histograms summarized their distribution. For clearer illustration, in this figure we instantiate the context distribution specifically by the node degree distribution. In the revised version, we have redrawn Figure 1, updated its caption, and added a clearer description of Figure 1 in lines 222-223.
>
> > W1b: Figure 2 has the same shortcomings as Figure 1 and likewise adds little to the paper; both read as fillers rather than explanatory figures.
>
> We keep Figure 2 because it provides a concrete example of a pair of graphs that are indistinguishable by 1-WL but can be distinguished by our method, and thus directly illustrates the gain in expressiveness. Additionally, we have added a detailed description to the caption of Figure 2 in the revised version. The explanation of the Figure is given in lines 243-248.
>
> > W1c & Q1: The theoretical statements are informal. For example, Theorem 2 begins “For this specific class of d-regular graphs with V nodes” but never defines the class. Theorem 3 mentions “significantly different node context distributions” informally and concludes that the method “can effectively reduce these differences,” which is also vague. Theorem 1 should be a proposition: by adding auxiliary information to a 1-WL GNN one can trivially subsume the 1-WL test and, as the 1-WL test equals the 2-WL, gnn can subsume it as well.
>
> We thank the reviewer for these helpful comments.
>
> First, following the reviewer’s suggestion, we have revised Theorem 1 into Proposition 1.
>
> Second, for Theorem 2, we apologize for the misunderstanding, this specific class only refers to $d$-regular graphs on $|V|$ vertices. To make the theoretical presentation more formal, we have also restated Theorem 2 (Theorem 1 in the revised version) as:
>
> **Theorem 1**: For the family $\mathcal{G}_{d,|V|}$ of all $d$-regular graphs with $|V|$ nodes, the ConD-MPNN with von Neumann entropy can distinguish at most $\mathcal{O}(|V|^2\log^2|V|)$ pairs of non-isomorphic regular graphs.
>
> Third, for the result previously presented as Theorem 3, we removed the informal phrases “significantly different” and “effectively reduce these differences” and restated it as a **precise inequality**, i.e., Proposition 2 in the revised version. Concretely, we now write as
>
> **Proposition 2**: Let $r_u$, $r_v$ denote the entropy-based representations of the node contexts,and $z_u$, $z_v$ be their kernel-based representation. Let $\sigma_1 = |r_u-r_v|$ and $\sigma_2 = |z_u-z_v| $, then, $\sigma_2<\sigma_1$.
>
> > W1d & Q3: It is unclear if the example in Figure 3 is representative or an outlier. Instead one would need to plot a measurement over multiple nodes. For Figure 3, how was the center node chosen and why 14 neighbors? Is this example representative or an outlier? Could you aggregate this analysis across many nodes/graphs (e.g., distribution of entropy spread before/after kernel mapping)?
>
> Thank you for your suggestions. In Figure 3, , the center node is randomly selected from a graph in the DD dataset and happens to have 14 neighbors, as determined by the graph structure. To verify that the observed effect is not accidental, we **now include additional visualization results over multiple nodes in Appendix 9.5**. As shown in Figure 3, the entropy-based representations exhibit the scale variation across nodes, with values spanning a wide range. In contrast, the kernel-based representations are more concentrated, demonstrating reduced inter-node discrepancy.

---

> ### Author Response · Authors · 2025-11-24
> **Response to Reviewer rgkQ (2)**
>
> > W2 & Q2: Unsupported efficiency claims. The assertion of “significantly reduced computational and memory costs” is not backed by end-to-end time and memory comparisons. The asymptotic discussion in the appendix is likewise unconvincing: since the subgraph size s in subgraph GNNs can be fixed as a constant, their asymptotic computational complexity matches that of the proposed method. The claim is therefore unsupported both asymptotically and empirically.
>
> Thank you for your valuable feedback. In the revised version, we have added a comparison analysis of end-to-end time and memory usage. We selected the COLLAB (sufficiently dense) and DD datasets (sufficiently large) as challenging settings. On these large graphs, we found that existing subgraph-based methods run out of memory on our hardware, which already reflects the higher efficiency and better scalability of our approach.
>
> For KP-GIN on COLLAB, the total time exceeds one day only for the preprocessing stage,  which further indicates that this method is not scalable to large datasets in practice. To provide a clearer quantitative contrast, we additionally report results on the medium-scale PROTEINS dataset, where our method still shows advantages in both running time and memory consumption.
>
> Although, in an asymptotic sense, the computational complexity of subgraph GNNs and general MPNNs is the same, in practice, it is still evident that they can reduce computation or memory access.
>
> | Model                | DD Total time (s) | DD Memory | COLLAB Total time (s) | COLLAB Memory | PROTEINS Total time (s) | PROTEINS Memory |
> | -------------------- | ----------------- | --------- | --------------------- | ------------- | ----------------------- | --------------- |
> | ESAN                 | OOM               | OOM       | OOM                   | OOM           | 1645                    | 2170M           |
> | GNN-AK+              | OOM               | OOM       | OOM                   | OOM           | 1821                    | 5468M           |
> | KP-GIN               | 19096             | **1116M** | $>>$1 day             | --            | 1960                    | 476M            |
> | **ConD-KGIN (ours)** | **7137**          | 1544M     | 35420                 | **2620M**     | **1317**                | **474M**        |
>
> > W3 & Q4: Reproducibility gaps. The reproducibility statement promises code in the supplementary materials, but no link is provided. Important hyperparameters and data-processing choices are missing or scattered across the appendix. This is especially concerning given the very strong BREC results.
>
> We thank the reviewer for carefully checking the reproducibility aspects. In the revised manuscript, we include an **anonymous link** in the *Reproducibility Statement* that provides the code and the corresponding TXT result files for the BREC dataset. For the hyperparameters of the BREC dataset, we set the context size to 4 and construct the context by selecting between random walk and shortest path methods. As for the combination of $\phi_{VN}(S_u^k)$ and $h_u$, we searched in {sum, concat}.
>
> > W4: Related work placement. The main paper lacks an in-depth comparison with competitive methods; a related-work section can be is found in the appendix, which reinforces the broader presentation issues.
>
> Thank you for your suggestion. In response, we have moved the Related Work section from the Appendix into the main paper (Section 2).

---

> ### Comment · Reviewer_rgkQ · 2025-11-25
>
> I thank the authors for their rebuttal, which partially addressed my concerns. I still find that the figures act more as placeholders than effective explanatory tools. Furthermore, while the BREC results are promising, the reliance on TUDatasets for real-world evaluation is a drawback, as these are widely considered poor benchmarks [1].
>
> I am raising my score to 4 to reflect the revisions, but I believe the paper still needs to improve its presentation and evaluation.
>
> [1] Position: Graph Learning Will Lose Relevance Due To Poor Benchmarks, ICML 2025

---

### Official Review · Reviewer_qzik · 2025-11-01

**Soundness:** 2
**Presentation:** 2
**Contribution:** 2
**Rating:** 4
**Confidence:** 4

**Summary:**

This paper proposes a GNN framework that propagates not only node features but also structural information of ego-subgraphs. The proposed framework can be broken down into the following key steps:

1. **Node Context Extraction**: For each node \(u\), the framework extracts its local context, denoted as \(S_u^k\), which represents the ego-subgraph centered at \(u\) and extending \(k\) hops outward. This ego-subgraph encompasses all nodes within \(k\) hops of \(u\) as well as the edges connecting them, effectively capturing the structural neighborhood around \(u\).

2. **Graph Entropy Computation**: The structural properties of each node's context \(S_u^k\) are encoded using graph entropy, specifically the von Neumann entropy. This computation produces a compact scalar representation that encapsulates the complexity and structural patterns of the ego-subgraph, providing a concise summary of its local topology.

3. **Entropy-Augmented MPNNs (ConD-MPNN)**: The computed entropy values are integrated into Message Passing Neural Networks (MPNNs) as supplementary features for neighboring nodes. This augmentation leads to the development of the ConD-MPNN model, which enhances both the expressiveness and computational efficiency of traditional MPNNs.

4. **Kernel-Based Structural Similarity (KerA-MPNN)**: To further enrich the framework, the entropy values of two ego-subgraphs are leveraged to compute a reproducing kernel between corresponding nodes. This kernel is incorporated into MPNNs as an additional edge feature, capturing structural similarities between the central node and its neighbors. This extension results in the KerA-MPNN model, which significantly improves the framework's ability to encode and utilize structural information.

**Strengths:**

1. Incorporating structural information of subgraphs into existing GNN models is a novel and promising approach to enhance model performance.
2. The paper provides formal results proving that the contextual distributional encoding (ConD-WL and ConD-MPNN) is strictly more expressive than 1-WL and 2-WL, and empirically distinguishes many regular graphs that are missed by 3-WL (Theorem 1, Theorem 2, and proofs in Appendix A.3, A.5).
3. Figures, such as Figure 1, cogently illustrate how the node context distributions (represented visually as colored histograms on graphs) are incorporated into message passing, offering clearer intuition than is common in related work.
4. The authors provide both theoretical and practical reasons for preferring von Neumann entropy over alternatives such as Shannon entropy, confirmed by discussion in Appendix A.4 and supporting experiments.
5. Reproducibility is well supported, with detailed methodology, extensive appendices, and code made available.

**Weaknesses:**

1. **Limited Novelty Relative to Advanced GNNs:** While the authors claim that the message passing scheme propagating both node features and contextual distributions is novel, it should be noted that message passing frameworks incorporating node features and structural information have been previously explored in the literature. For instance, [4] presents a similar approach where structural information is integrated with node features in classic GNN architectures. Thus, the claim that this framework represents a significant shift from operations on node-tuples or subgraphs to the structural distribution of node contexts appears overstated.
2. **Insufficient Theoretical Validation of the Reproducing Kernel:** While the authors claim that the proposed context entropy reproducing kernel captures similarities between different structural distributions, they do not provide rigorous mathematical proof that this kernel actually encodes meaningful structural similarities. The theoretical analysis is limited to Theorem 3, which focuses on reducing discrepancies rather than formally establishing the kernel's ability to capture structural similarities. A comprehensive mathematical analysis would require a thorough examination of the kernel's properties and its relationship with the underlying structural distributions. Such rigorous theoretical validation is essential for acceptance in top-tier AI conferences.
3. **Insufficient Theoretical and Empirical Validation of Expressivity:** While the paper compares ConD-MPNN against 1-WL, 2-WL, and 3-WL tests, it lacks theoretical and empirical comparison with k-hop MPNNs. Since the evaluation relies heavily on k-hop MPNNs with structural information, a theoretical comparison with k-hop MPNNs is needed to properly assess the expressivity gains. Furthermore, empirical results of the native k-hop MPNNs on BREC, QM9, ZINC, and Peptides datasets are needed. Without these baselines, the improvement achieved by the proposed methods over standard k-hop MPNNs cannot be quantified.
4. **There are several critical issues in the procedure for constructing the $k$-hop neighborhood:**
    1. The phrase `random walk distance' lacks a widely accepted or standardized definition, potentially causing confusion and ambiguity.
    2. Line 7 is redundant since setting $A_{\text{final}}[A_{\text{final}} > 0] \gets 0$ always results in a zero matrix, negating its intended purpose. Consequently, the operation in line 8 simplifies to $A_{\text{final}} = A^i$. As a result, $A_{\text{final}}[u,v] > 0$ fails to indicate whether the shortest-path distance between nodes $u$ and $v$ equals $i$.
    3. Lines 10-12 should be moved outside the for loop (line 2) since the algorithm returns the context extracted from the fully accumulated neighborhood, making their current placement within the loop redundant.
    4. Line 5, $A_{\text{final}} \gets A_{\text{final}} + A^i$, ensures $A_{\text{final}}[u,v]>0$ when the shortest path distance between $u$ and $v$ is $\leq k$. Thus, the 'random walk' method actually extracts node contexts based on shortest path distances.
5. The main text lacks a clear introduction to the key backbone model, the K-hop MPNN framework, which makes it difficult to fully understand the proposed framework and its components.
6. Two backbones are both named K-GIN: one from (Morris et al., 2019) and another from (Nikolentzos et al., 2020). This shared naming confuses readers. Especially, it is unclear which K-GIN variant is used for ConD-KGIN and KerA-KGIN in Table 4, 5 and 10. In addition, Both K-GIN (Nikolentzos et al., 2020) and KP-GIN (Feng et al., 2022) are k-hop MPNNs. The authors do not clearly distinguish between K-GIN and KP-GIN before the comparison, which may confuse readers.
7. In line 355, the authors claim that both ConD-based and KerA-based MPNNs can distinguish graphs that are not distinguishable by the 4-WL test. However, there is insufficient theoretical or comprehensive empirical evidence provided to substantiate this strong claim.
8. The improvements achieved by the proposed methods on TU datasets are limited.
9. The evaluation on QM9 and ZINC lacks comparison with recent SOTA models such as PPGN+RRWP[1], N$^2$-GNN[2], and GRIT[3]. The proposed methods perform worse than these SOTA models, particularly PPGN+RRWP, failing to demonstrate superiority over recent SOTA models.

**Questions:**

1. How does the expressive power of the proposed graph-entropy encoding compare to the k‑FWL hierarchy?
2. How does ConD‑KGIN’s expressivity compare to existing k‑hop MPNNs?

---

> ### Author Response · Authors · 2025-11-24
> **Response to Reviewer qzik (1)**
>
> Thank you for your valuable comments! In the following section, we address the weaknesses (W) and questions (Q) mentioned above. **The changes in the revised version are marked in blue.**
>
> > W1: Limited Novelty Relative to Advanced GNNs
>
> R1: We thank the reviewer for this valuable feedback, which helped us realize that the novelty of our method was not clearly presented in the original version. We agree that there exists a rich line of work that incorporates structural information into message passing [4, 5]. In the revised manuscript, we have therefore toned down the wording in the abstract and introduction and now position our framework as providing a novel paragdim rather than a “significant shift”.
>
> At the same time, we would like to clarify more precisely how our approach differs from existing structural message passing schemes. Compared to prior work, our method explicitly models the **distribution of $k$-hop contexts** around each node and propagates a compressed representation of this distribution via **entropy** (structural Shannon entropy or von Neumann entropy). This context-distribution view (i) does not rely on a predefined set of substructures, (ii) operates directly at the distribution level, and (iii) can be used as a plug-in for various backbones while improving expressive power on isomorphism benchmarks.
>
> Additionally, we achieve SOTA results in expressiveness evaluations, e.g., Table 2 on the BREC dataset as below, which empirically support that focusing on context distributions and their entropy properties enhances the ability to distinguish non-isomorphic graphs and leads us to explore the relationship between expressiveness and the distribution of eigenvalues.
>
> | Method  | Basic (60) | Regular (140) | Extension graph (100) | CFI (100) | Total (400) | Accuracy   |
> | ------ | -- | -- | --| -- | --| -- |
> | I²-GNN   | 60         | 100           | **100**  | 21        | 281         | 70.2%      |
> | GNN-AK+   | 60         | 50            | 97        | 15        | 222         | 55.5%      |
> | PPGN  | 60         | 50            | 100          | 23        | 233         | 58.2%      |
> | NGNN    | 59         | 48            | 59          | 0         | 166         | 41.5%      |
> | GRIT     | 60         | 50            | 100           | 16        | 226         | 56.5%      |
> | N²GNN    | 60         | 100           | 100       | 27        | 287         | 71.8%      |
> | DSS-GNN   | 58         | 48     | **100**  | 15        | 221         | 55.2%      |
> | K-GIN     | 60         | 48            | 100       | 27        | 235         | 58.8%      |
> | KP-GIN    | 60         | 106           | 98       | 11        | 275         | 68.8%      |
> | **ConD-KGIN (ours)** | **60**     | **100**       | 98     | **78**    | **336**     | **84%**    |
> | **KerA-KGIN (ours)** | **60**     | **118**       | 81     | **72**    | **331**     | **82.75%** |
>
> > W2:  Insufficient Theoretical Validation of the Reproducing Kernel.
>
> A2: We thank the reviewer for this insightful comment. First, we would like to clarify that our explanation is based on the definition of kernel methods, which is grounded in the established principle of capturing similarity, as mentioned in reference [1]: "the similarity measure $ k$ is usually called a kernel."  The reviewer’s comment has made us realize the need for a more formal proof. In response, we have added a more systematic analysis of the proposed kernel, and a clearer, more detailed version of the proof is highlighted in Appendix A.6 of the revised version.
>
> Intuitively, our kernel
> $$
> \mathcal{K}(x,y) = e^{-\lVert x - y \rVert}
> $$
> is a bounded, symmetric function that is **monotonically decreasing in the entropy distance** between $x$ and $y$. In our setting, $x$ and $y$ are entropy-based representations of the structural distributions of the contexts $S_u^k$ and $S_v^k$. Thus, when two node contexts have identical structural distributions, their entropy representations are the same and then $\mathcal{K}(x,y)=1$; as their structural distributions diverge, the entropy distance $\lVert x-y \rVert$ increases and the kernel value gradually drops toward 0. Thus, similar structural distributions produce large kernel values, while dissimilar ones produce small kernel values, which is precisely the notion of structural similarity that we aim to capture.
>
> From a more rigorous mathematical standpoint, the kernel can be written as the inner product of two feature mappings, which makes its role in capturing similarity more explicit.
> For the reproducing kernel $ \mathcal{K}(x, y) = e^{-|x-y|}$, there exists an explicit feature mapping $\phi$ defined by
> $$
>   \Phi(x) = ( \sqrt{\frac{2}{\pi(1+\omega^2)}} \cos(\omega x),\ \sqrt{\frac{2}{\pi(1+\omega^2)}} \sin(\omega x) )_{\omega \geq 0},
> $$
>  We  can verify the validity by computing the inner product integral:
> $$
> \langle \Phi(x), \Phi(y) \rangle = e^{-|x-y|}
> $$
> The detailed proof is shown in Appendix A.6 of the revised version.

---

> ### Author Response · Authors · 2025-11-24
> **Response to Reviewer qzik (2)**
>
> > W3:  Insufficient Theoretical and Empirical Validation of Expressivity
>
> R3: We thank the reviewer for pointing this out, and we have addressed this in the revised version.
>
> **Empirical comparison with k-hop MPNNs.**
>
> We have reproduced the k-hop MPNN (K-GIN) and reported its experimental results on the BREC, QM9, Peptides, and ZINC datasets. The tables below compare the performance of K-GIN and our ConD-KGIN on these benchmarks.
>
> On the BREC dataset:
>
> | Model  | Basic | Regular | Extension graph | CFI  | Total | Accuracy  |
> | -- | :---: | :-----: | :-------------: | :--: | :---: | :-------: |
> | K-GIN       |  60   |   48    |  100       |  27  |  235  |   58.8%   |
> | **ConD-KGIN (ours)** |  60   |   100   |       98        |  78  |  336  | **84.0%** |
>
> On the QM9 dataset:
>
> | Target | K-GIN   | KP-GIN  | **ConD-KGIN** |
> | ---- | --- | - | --|
> | $\mu$  | 0.430   | 0.358   | 0.360         |
> | $\alpha$    | 0.310   | 0.233   | **0.224**     |
> | $\epsilon_{HOMO}$     | 0.00301 | 0.00240 | **0.00237**   |
> | $\epsilon_{LUMO}$     | 0.00288 | 0.00236 | **0.00228**   |
> | $\Delta \epsilon$     | 0.00416 | 0.00333 | **0.00330**   |
> | $\langle R^2 \rangle$ | 23.37   | 16.51   | **16.26**     |
> | ZPVE   | 0.00033 | 0.00017 | 0.00018       |
> | $U_0$   | 0.0602  | 0.0682  | **0.0667**    |
> | $U$       | 0.0504  | 0.0696  | **0.0323**    |
> | $H$      | 0.0534  | 0.0641  | **0.0344**    |
> | $G$          | 0.0533  | 0.0484  | **0.0309**    |
> | $C_\nu$      | 0.1365  | 0.0869  | 0.0916        |
>
> On the Peptides dataset:
>
> | Method  | Peptides (Test AP ↑) |
> | ---- | -- |
> | K-GIN    | 0.6479 ± 0.0041 |
> | **ConD-KGIN (ours)** | **0.6901 ± 0.0034**  |
>
> On the ZINC dataset:
>
> | Method     | Time (s) | Params  | MAE (↓)  |
> | - | -- | -- | -- |
> | K-GIN | 8.77     | 487465     | 0.097 ± 0.004     |
> | **ConD-KGIN** | **7.33** | **487465** | **0.086 ± 0.007** |
>
> **Theoretical comparison with k-hop MPNNs.**
>
> Theoretically, ConD-MPNN can be viewed as a general plugin that can be applied to all MPNN frameworks. Our core contribution is the introduction of this plugin, which enhances the expressiveness of the original framework, with K-MPNN being one of the frameworks. As pointed out in the GIN[2], MPNNs can only match the expressiveness of 1-WL in the best case. By incorporating the proposed plugin, the model surpasses 1-WL in terms of expressiveness. Therefore, we have shown that the expressiveness of ConD-MPNN exceeds that of traditional MPNNs, and it directly follows that the expressiveness of ConD-KMPNN is greater than that of K-MPNN.
> > W4: **There are several critical issues in the procedure for constructing the $k$-hop neighborhood**
>
> R4: We sincerely appreciate the reviewer’s careful evaluation and apologize for the confusion caused. In the revised version, we have updated the pseudocode in Appendix A.1.
>
> 1&4: Regarding points 1 and 4, we compute the length of the random walk, i.e., $A^i$. Ultimately, we not only extract the nodes but also return the adjacency matrix, which is used to extract the distances from the center node to the k-hop neighboring nodes, serving as edge attributes for the k-hop MPNN. It is important to note that the length of random walks and shortest paths are different.
>
> 2. The original line 5 should be $A^i[A_{final}>0] = 0$, which is used to mask nodes already present in the previous hops for extracting the shortest path.
>
> 3. The final output should be the context of the 1 to k hops, $S_u^{1:k}$, so the loop should be placed inside.
> The updated pseudocode is shown below:
>
> **Algorithm 1** Algorithm for extracting node context based on $(1–k)$-hop neighborhood
>
> **Input:** Graph $G$, adjacency matrix $A$, number of hops $k$, number of nodes $|V|$, method type (random walk or shortest path )
>
> **Output:** Node contexts based on the $(1-k)$-hop neighborhood $S_u^{1:k}$, edge attributes for $k$-hop MPNN (the list of adjacency matrices $A_{\text{list}}$)
>
>
> 1. Initialize an empty adjacency matrix $A_{\text{final}}$.
>
> 2. Initialize a list $A_{\text{list}}$ of length $k$, where the $i$-th element will store the adjacency matrix of the $i$-hop neighbors.
>
> 3. **For** $i = 1$ to $k$ **do**
>
>    3.1 Compute the $i$-th power of the adjacency matrix:
>     $A^{(i)} = A^i$
>
>    3.2 **If** method == "random walk" **then**
>     a) Update $A_{\text{final}} \gets A_{\text{final}} + A^{(i)}$
>     b) Set $A_{\text{list}}[i] \gets A^{(i)}$
>
>    3.3 **Else if** method == "shortest path" **then**
>     a) Mask nodes already existing in previous hops:
>     $A^{(i)}[A_{\text{final}} > 0] \gets 0$
>     b) Update $A_{\text{final}} \gets A_{\text{final}} + A^{(i)}$
>     c) Set $A_{\text{list}}[i] \gets A^{(i)}$
>
>    3.4 **For each** node $u \in G$ **do**
>     a) Extract neighbors $N_u^{1:i}$ such that $A_{\text{final}}[u, v] > 0$ for $v \in N_u^{1:i}$
>     b) Update the node context:
>     $S_u^i = A_{\text{final}}[N_u^{1:i}][:, N_u^{1:i}]$
>
>    3.5 **End for**
>
> 4. **Return** the $(1-k)$-hop node contexts $S_u^{1:k}$ and the list $A_{\text{list}}$.

---

> ### Author Response · Authors · 2025-11-24
> **Response to Reviewer qzik (3)**
>
> > W5: The main text lacks a clear introduction to the key backbone model, the K-hop MPNN framework, which makes it difficult to fully understand the proposed framework and its components.
>
> R5:  We thank the reviewer for pointing this out. In the revised version, we have
>
> - added a more detailed introduction to K-hop MPNN in the Related Work section,
> - included the explicit formulation of K-hop message passing neural network in the Preliminary Concepts section.
>
> > W6: It is unclear which K-GIN variant is used for ConD-KGIN and KerA-KGIN in Table 4, 5 and 10. In addition, Both K-GIN (Nikolentzos et al., 2020) and KP-GIN (Feng et al., 2022) are k-hop MPNNs. The authors do not clearly distinguish between K-GIN and KP-GIN before the comparison, which may confuse readers.
>
> R6:  We thank the reviewer for pointing out this source of confusion and we apologize for not making this clearer in the original version.
>
> First, we clarify that in **all** our experiments, the backbones for **ConD-KGIN** and **KerA-KGIN** in Tables 4, 5, and 10 are based on **K-GIN as proposed by Nikolentzos et al. (2020)**. We agree with the reviewer that the K-GIN (Morris et al., 2019) is different from K-GIN (Nikolentzos et al., 2020), as discussed in the related work, there are also higher-order GNNs inspired by k-WL(K-GIN (Morris et al., 2019)), where message passing is performed between K-tuples, whereas K-GIN (Nikolentzos et al., 2020) aggregates information from the K-hop neighborhood of nodes. Therefore, all of our work is based on the latter framework.  As stated in line 299-300 of our paper, “we extend the proposed ConD-MPNN to the K-hop MPNN framework (Nikolentzos et al., 2020).”
>
> Second, regarding the difference between KP-GIN and K-GIN, K-hop Peripheral-subgraph-enhanced (KP-GIN) (Feng et al., 2022) is based on the K-GIN (Nikolentzos et al., 2020) framework, with the addition of K-hop Peripheral Subgraph information. Therefore, in our manuscript, K-GIN and KP-GIN therefore refer to different methods.
>
> > W7:  In line 355, the authors claim that both ConD-based and KerA-based MPNNs can distinguish graphs that are not distinguishable by the 4-WL test. However, there is insufficient theoretical or comprehensive empirical evidence provided to substantiate this strong claim.
>
> R7: We thank the reviewer for pointing out that our original statement was too strong. In our experiments, we provided results on the BREC dataset, which includes graphs that cannot be distinguished by 4-WL (e.g., CFI), our proposed method can distinguish over 70% of these graphs. However, we would like to clarify that it does not mean we can distinguish all graphs that 4-WL cannot distinguish. Therefore, we have revised the original text in Line 420 of the updated version to avoid making overly absolute statements. In fact, designing a computationally efficient method that can separate all 4-WL-indistinguishable graphs remains a challenging open problem. Besides, our approach already achieves **state-of-the-art performance on BREC** among existing GNN-based methods.
>
> > W8: The improvements achieved by the proposed methods on TU datasets are limited.
>
> R8: We understand that the improvements on TU datasets are relatively modest.This is expected, as these small-scale graph classification benchmarks are known to be saturated and do not strongly stress expressive power. Our main contribution lies in the design of a plug-and-play module that primarily enhances **the expressiveness of GNNs**, i.e., its ability to distinguish non-isomorphic graphs, rather than pushing the SOTA on TU classification benchmarks. As shown in the experimental results in Table 1& 2, especially for BREC dataset below, our method achieves SOTA performance in terms of expressiveness. At the same time, when plugged into different backbones, our methods still achieve **competitive** and often improved performance in graph classification and graph regression tasks, even though they are not always strictly SOTA.
>
> | Method   | Basic (60) | Regular (140) | Extension graph (100) | CFI (100) | Total (400) | Accuracy   |
> | ----- | ---- | ---- | --------------------- | --------- | ----------- | ---------- |
> | I²-GNN      | 60         | 100           | **100**               | 21        | 281         | 70.2%      |
> | GNN-AK+      | 60         | 50            | 97      | 15        | 222         | 55.5%      |
> | NGNN         | 59         | 48            | 59       | 0         | 166         | 41.5%      |
> | DSS-GNN    | 58         | 48            | **100**     | 15        | 221         | 55.2%      |
> | K-GIN     | 60         | 48            | 100      | 27        | 235         | 58.8%      |
> | KP-GIN               | 60         | 106           | 98   | 11        | 275         | 68.8%      |
> | **ConD-KGIN (ours)** | **60**     | **100**       | 98     | **78**    | **336**     | **84%**    |
> | **KerA-KGIN (ours)** | **60**     | **118**       | 81      | **72**    | **331**     | **82.75%** |

---

> ### Author Response · Authors · 2025-11-24
> **Response to Reviewer qzik (4)**
>
> > W9: The evaluation on QM9 and ZINC lacks comparison with recent SOTA models such as PPGN+RRWP[1], N$^2$-GNN[2], and GRIT[3]. The proposed methods perform worse than these SOTA models, particularly PPGN+RRWP, failing to demonstrate superiority over recent SOTA models.
>
> R9: We agree that our methods do not surpass the very recent SOTA models such as PPGN+RRWP, N$^2$-GNN, and GRIT on QM9 and ZINC. As mentioned in our previous response, our core contribution is not to achieve the SOTA in graph regression tasks, but rather to design a plug-and-play module that enhances expressiveness. In these downstream tasks, we believe that our method already demonstrates competitive performance compared to existing subgraph-based methods. More importantly, our method achieves SOTA performance on the BREC dataset as below, which is specifically designed to evaluate the expressiveness. Following the reviewer’s suggestion, we have additionally included the performance of PPGN, N$^2$-GNN, and GRIT on BREC in the revised manuscript. As shown in the table below, our ConD-KGIN and KerA-KGIN clearly outperform these models in terms of distinguishing hard non-isomorphic graph pairs.
>
> | Method               | Basic (60) | Regular (140) | Extension graph (100) | CFI (100) | Total (400) | Accuracy   |
> | -------------------- | ---------- | ------------- | --------------------- | --------- | ----------- | ---------- |
> | I²-GNN               | 60         | 100           | **100**               | 21        | 281         | 70.2%      |
> | GNN-AK+              | 60         | 50            | 97                    | 15        | 222         | 55.5%      |
> | **PPGN**                 | 60         | 50            | 100                   | 23        | 233         | 58.2%      |
> | NGNN                 | 59         | 48            | 59                    | 0         | 166         | 41.5%      |
> | **GRIT**                 | 60         | 50            | 100                   | 16        | 226         | 56.5%      |
> | **N²GNN**                | 60         | 100           | 100                   | 27        | 287         | 71.8%      |
> | DSS-GNN              | 58         | 48            | **100**               | 15        | 221         | 55.2%      |
> | K-GIN                | 60         | 48            | 100                   | 27        | 235         | 58.8%      |
> | KP-GIN               | 60         | 106           | 98                    | 11        | 275         | 68.8%      |
> | **ConD-KGIN (ours)** | **60**     | **100**       | 98                    | **78**    | **336**     | **84%**    |
> | **KerA-KGIN (ours)** | **60**     | **118**       | 81                    | **72**    | **331**     | **82.75%** |
>
> > Q1: How does the expressive power of the proposed graph-entropy encoding compare to the k‑FWL hierarchy?
>
> A1: Thank you for your insightful question. As proven in reference [3], (k-1)-FWL = k-WL, which means that comparing our method to k-WL in the paper is effectively equivalent to comparing it to the k-FWL hierarchy.
>
> > Q2: How does ConD‑KGIN’s expressivity compare to existing k‑hop MPNNs?
>
> A2: To provide a clearer empirical comparison, we have added the results of K-GIN on the BREC dataset in the revised version. As shown in the table below, which is also updated in Table 2 of the revised version. As can be seen, **ConD-KGIN** achieves a substantially higher total score and accuracy, especially on the challenging CFI instances.
>
> | Model     | Basic | Regular | Extension graph | CFI | Total | Accuracy |
> |---------------|-------|---------|-----------------|-----|-------|----------|
> | K-GIN     | 60   | 48    | 100       | 27  | 235  | 58.8%   |
> | ConD-KGIN (ours)   | 60   | 100   | 98        | 78  | 336  | 84%    |
>
> Since BREC is specifically designed to evaluate **expressive power**, these results indicate that ConD-KGIN exhibits **stronger expressivity than existing K-hop MPNNs on this benchmark**. For the theoretical comparison with standard MPNNs and its extension to k-hop settings, please refer to our reply to W3.
>
> [1] Hofmann T, Schölkopf B, Smola A J. Kernel methods in machine learning[J]. 2008.
>
> [2] Keyulu Xu, Weihua Hu, Jure Leskovec, and Stefanie Jegelka. How powerful are graph neural networks? In Proceedings of ICLR, 2019, New Orleans, LA, USA, May 6-9, 2019.
>
> [3] Huang, N. T. and Villar, S. A short tutorial on the weisfeilerlehman test and its variants. In ICASSP 2021-2021. IEEE International Conference on Acoustics, Speech and Signal Processing (ICASSP), pp. 8533–8537. IEEE, 2021.
>
> [4] Kanatsoulis et al., Learning Efficient Positional Encodings with Graph Neural Networks. In Proceedings of ICLR, 2025
>
> [5] Bouritsas G, Frasca F, Zafeiriou S, et al. Improving graph neural network expressivity via subgraph isomorphism counting[J]. IEEE Transactions on Pattern Analysis and Machine Intelligence, 2022, 45(1): 657-668.

---

> > ### Comment · Reviewer_qzik · 2025-11-25
> >
> > The novelty of the proposed method relative to existing advanced approaches remains unclear and appears limited.
> >
> > The approach models the distribution of k-hop neighborhoods around each node and summarizes it using entropy as a form of structural encoding. However, the paper offers neither a theoretical analysis nor empirical comparisons against well-established structural encodings, such as RRWP [1] and Cheb [1], making it difficult to assess the added value of the proposed technique.
> >
> > More critically, the entropy measure is derived from the Laplacian eigenvalues. Yet, prior work [1] has already established a theoretical upper bound on the expressivity of Laplacian-based encodings and introduced Cheb as a method that empirically achieves this bound. Consequently, the proposed method offers no clear theoretical or practical advantage over existing approaches.
> >
> > While the authors report SOTA results by combining their encoding with kGIN, this alone does not demonstrate the superiority of their encoding. A rigorous evaluation would require direct comparisons between kGIN + the proposed encoding and kGIN + other advanced structural encodings (e.g., RRWP, Cheb). Without such ablation studies, the claimed contribution remains unsubstantiated.
> >
> > As the core concerns about the method's novelty remain unaddressed, and without implying that other concerns have been resolved, I continue to believe the paper falls short of the threshold for acceptance.
> >
> > [1] Enhanced Subgraph Learning in 2-FWL GNNs via Local Connectivity, Spectral, and Distance Encodings.

---

### Official Review · Reviewer_uqWp · 2025-11-01

**Soundness:** 2
**Presentation:** 1
**Contribution:** 2
**Rating:** 2
**Confidence:** 3

**Summary:**

The paper proposes to incorporate structural information in the message passing mechanism; the structural information is represented by the von Neumann entropy of the k-hop subgraph of the target node. The results show improved performance when this structural information is encoded in message passing.

**Strengths:**

- The use of von Neumann entropy metric to incorporate structural information is a novel and interesting exploration in message passing on graphs.

**Weaknesses:**

- The presentation is unclear, and the entire paper is difficult to understand. For example, the functional forms of **U** and **M** in equations 7, 8, and 9 are not given. This makes it difficult to understand the proposed method. Also, what is the dimensionality of the linear transformation $\phi$?

- It is difficult to conclude that von Neumann entropy is better than Shannon entropy from Table 16. The performances are not statistically different considering the standard deviation.

- The result of the proposed methods in Table 3 compared to GIN and K-GIN are not statistically different. This does not validate the effectiveness of the proposed message passing strategy. Also, the extension to GCN and GraphSAGE backbones in Table 9 has no significant improvement over the baseline.

- It is not clear why K-GIN and KP-GIN are excluded from the baselines in Table 2.

- The kernel-based structural similarity does not seem to bring any advantage compared to the ConD variant; the performances are not statistically different between the two methods.

I recommend that the authors substantially revise the paper to improve the mathematical clarity of the proposed methods and to further investigate why incorporating structural information does not yield meaningful performance gains. The current results directly challenge the core motivation behind injecting additional structural signals for representation learning: the results suggest that message passing schemes in standard architectures such as GCN and GraphSAGE may already implicitly capture the relevant structural information, and that explicitly adding more structure offers no tangible benefit in cases os benchmark or real-world graphs.

**Questions:**

- It is not clear why "kernel-based aggregation strategy effectively mitigating the discrepancies in node context distributions" leads to better performance.
- In Figure 3 (a), does the entropy value depend on the central node?

---

> ### Author Response · Authors · 2025-11-24
> **Response to Reviewer uqWp (1)**
>
> Thank you for your valuable comments! In the following section, we address the weaknesses (W) and questions (Q) mentioned above. **The changes in the revised version are marked in blue.**
>
> > W1: The presentation is unclear, and the entire paper is difficult to understand. For example, the functional forms of **U** and **M** in equations 7, 8, and 9 are not given. This makes it difficult to understand the proposed method. Also, what is the dimensionality of the linear transformation
>
> R1: We thank the reviewer for pointing out this issue. In response to your suggestion, we have substantially revised the paper to improve the presentation and overall readability.
>
> For $\mathbf{M}^t$ and $\mathbf{U}^t$, the definitions are given in Eq. (3), which was located in Section 2 of the original version and has now been moved to Section 3 in the revised manuscript. Equation (3) provides a **general formulation** of message passing neural networks,, where $ \mathbf{M}^t $ and $ \mathbf{U}^t $ denote the message and update functions at iteration $ t $. Since our method is designed as a plug-in that can be integrated into generic message passing networks, the detailed forms of $\mathbf{M}^t$ and $\mathbf{U}^t$ depend on the chosen backbone.
>
> To make this clearer, in the revised manuscript we now explicitly state around Eqs. (7)–(9) that $\mathbf{M}^t$ and $\mathbf{U}^t$ follow the general definition in Eq. (3), and we provide a concrete example using the GIN backbone in Eq. (5), in this case, $\mathbf{M}^t$ corresponds to the summation over neighboring node features and $\mathbf{U}^t$ is instantiated as an MLP. We also add a cross-reference to Eq. (5) in Eq (9) so that readers do not need to infer these forms implicitly.
>
> Regarding the dimensionality of the linear transformation, we now clarify that this transformation is applied to the plug-in features, and its output dimension is chosen to **match the hidden feature dimension** of the backbone GNN so that it can be added to the hidden node representations. For example, for ConD-GIN and KerA-GIN, the dimensionality is selected from $\{16, 32, 64, 128\}$. For ConD-KGIN and KerA-KGIN, the dimensionality is set to 128 on QM9, and is selected from $\{94, 106\}$ on ZINC. These choices are now explicitly described in the implementation details in Appendix A9.2 of the revised version.
>
> > W2: It is difficult to conclude that von Neumann entropy is better than Shannon entropy from Table 16. The performances are not statistically different considering the standard deviation.
>
> R2: We appreciate the reviewer’s careful examination of Table 16 and agree with this assessment. However, our intention was not to claim a strong overall superiority of von Neumann entropy on the graph classification task. In the revised manuscript, we have toned down the corresponding statements and now describe as：
>
> "The results are showing that both von Neumann entropy and structural Shannon entropy are effective components within our framework and achieve comparable performance on graph classification tasks, **with von Neumann entropy exhibiting a slight advantage in terms of mean accuracy and smaller variance**."
>
> At the same time, one of our main motivations for introducing von Neumann entropy is its **stronger ability to distinguish non-isomorphic graphs**. To better support this point, we have added the comparison on three expressive-power evaluation datasets in the revised version. The results below show that the von Neumann entropy consistently outperforms structural Shannon entropy.
>
> | Method                     |    EXP    |    CSL    |    SR     |
> | -------------------------- | :-------: | :-------: | :-------: |
> | Structural Shannon Entropy | **100.0** |   26.67   |   96.7    |
> | **von Neumann entropy**    | **100.0** | **100.0** | **100.0** |

---

> ### Author Response · Authors · 2025-11-24
> **Response to Reviewer uqWp (2)**
>
> > W3: The result of the proposed methods in Table 3 compared to GIN and K-GIN are not statistically different. This does not validate the effectiveness of the proposed message passing strategy. Also, the extension to GCN and GraphSAGE backbones in Table 9 has no significant improvement over the baseline.
>
> R3: We agree that, given the reported means and standard deviations, the performance gains on standard graph classification benchmarks are relatively modest. However, following standard practice in graph learning, we primarily compare mean performance across folds/runs, while the standard deviation is reported to indicate stability across data splits , i.e., smaller variance is preferable when the mean performance is similar.
>
> Moreover, our intention in these Tables is not to argue that our message passing strategy leads to large performance jumps, but to demonstrate that the proposed plug-in frameworks (ConD- and KerA-) can be integrated into different backbones (GIN, K-GIN, GCN, and GraphSAGE) without hurting performance and often with consistent improvements. For example,  as shown in the table below, on some datasets such as MUTAG, PTC, IMDB-BINARY, and DD, ConD-GIN and KerA-GIN achieve higher mean accuracy than GIN, and ConD-KGIN / KerA-KGIN improve over K-GIN on most datasets.  For the extension to GCN and GraphSAGE backbones in Table 9, our variants match or slightly exceed the performance of the strong baselines KP-GCN and KP-GraphSAGE on several benchmarks, rather than only improving over the original backbones.
>
> | Method        | MUTAG          | PTC            | PROTEINS       | IMDB-BINARY    | IMDB-MULTI     | DD             | COLLAB         |
> | ------------- | -------------- | -------------- | -------------- | -------------- | -------------- | -------------- | -------------- |
> | GIN           | 89.4 ± 5.6     | 64.6 ± 7.0     | 75.9 ± 2.8     | 75.1 ± 5.1     | 52.3 ± 2.8     | 75.3 ± 2.9     | 80.2 ± 1.9     |
> | **ConD-GIN**  | **89.9 ± 5.0** | **66.0 ± 6.5** | **76.0 ± 3.9** | **76.3 ± 4.3** | **53.2 ± 3.4** | **78.9 ± 3.5** | **80.5 ± 1.7** |
> | **KerA-GIN**  | **91.5 ± 5.4** | **66.9 ± 7.5** | **76.4 ± 4.4** | **75.3 ± 2.7** | **53.1 ± 3.6** | **79.4 ± 4.4** | **80.6 ± 2.1** |
> | K-GIN         | 88.9 ± 8.3     | 63.8 ± 8.8     | 76.0 ± 5.0     | 73.0 ± 3.5     | 52.7 ± 3.7     | 79.5 ± 6.6     | 80.8 ± 2.1     |
> | **ConD-KGIN** | **93.9 ± 5.5** | **67.4 ± 9.4** | **77.0 ± 3.1** | **76.8 ± 1.9** | **53.3 ± 2.5** | **79.7 ± 3.0** | **82.7 ± 1.2** |
> | **KerA-KGIN** | **92.8 ± 4.6** | **67.6 ± 6.4** | **76.7 ± 6.5** | **76.6 ± 4.1** | **53.1 ± 3.2** | **80.4 ± 4.9** | **82.4 ± 2.1** |
>
> We thank the reviewer for these comments and have revised the wording in the manuscript accordingly to describe these results as showing **competitive or slightly better performance** than the baselines.
>
> > W4: It is not clear why K-GIN and KP-GIN are excluded from the baselines in Table 2.
>
> R4: We thank the reviewer for pointing this out. Table 2 focuses on the expressiveness evaluation on the BREC dataset, and in the original submission we prioritized methods that are state-of-the-art or representative in terms of expressive power on BREC. We agree, however, that including K-GIN and KP-GIN as additional baselines makes the comparison more complete. Following the reviewer’s suggestion, we have now added both K-GIN and KP-GIN to Table 2 in the revised manuscript. We reproduce the results of K-GIN on the BREC dataset using the authors’ implementation, and we include the KP-GIN results reported in [1]. The updated results are:
>
> | Model                | Basic | Regular | Extension graph | CFI  | Total | Accuracy  |
> | -------------------- | :---: | :-----: | :-------------: | :--: | :---: | :-------: |
> | K-GIN                |  60   |   48    |       100       |  27  |  235  |   58.8%   |
> | KP-GIN               |  60   |   106   |       98        |  11  |  275  |   68.8%   |
> | **ConD-KGIN (ours)** |  60   |   100   |       98        |  78  |  336  | **84.0%** |
>
> As shown above, our ConD-KGIN achieves a higher total accuracy than both K-GIN and KP-GIN, so the main conclusions of the paper remain unchanged. The revised Table 2 in the manuscript has been updated accordingly.

---

> ### Author Response · Authors · 2025-11-24
> **Response to Reviewer uqWp (3)**
>
> > W5:  The kernel-based structural similarity does not seem to bring any advantage compared to the ConD variant
>
> R5:would like to clarify that, our goal with the KerA variant is not to replace ConD, but to provide a complementary way of aggregating structural information by explicitly modeling **kernel-based similarities between context distributions**. In practice, on some tasks or datasets, KerA variant does not degrade performance and often yields slightly better results. For example, within the GIN framework, KerA-GIN achieves higher mean accuracy than ConD-GIN on several datasets, as shown below:
>
> | Method   | MUTAG    | PTC  | PROTEINS   | DD    | COLLAB  |
> | --- | -- | -- | -- | -- | --- |
> | ConD-GIN | 89.9 ± 5.0 | 66.0 ± 6.5 | 76.0 ± 3.9 | 78.9 ± 3.5 | 80.5 ± 1.7 |
> | KerA-GIN | 91.5 ± 5.4 | 66.9 ± 7.5 | 76.4 ± 4.4 | 79.4 ± 4.4 | 80.6 ± 2.1 |
>
> Moreover, on certain regression targets in QM9, the KerA variant shows a clearer advantage over ConD-KGIN, as summarized below (lower is better):
>
> | Target | ConD-KGIN | KerA-KGIN |
> | - | - | --|
> | **ε\_LUMO** | 0.00228   | 0.00228  |
> | **ZPVE**    | 0.00018   | 0.00017  |
> | **H**       | 0.0344    | 0.0268    |
> | **C\_v**    | 0.0916    | 0.0899    |
>
> These results suggest that KerA provides a **kernel-based aggregation mechanism** that is at least as competitive as ConD on standard benchmarks, and can be more advantageous on some tasks and targets, especially those that are sensitive to finer-grained differences in context distributions. In the revised manuscript, we have added additional visualizations in Figure 5 to further illustrate this behavior. Overall, we conclude that both ConD and KerA are effective instantiations of our framework, with KerA offering a principled kernel-based alternative that can lead to slight improvements in several cases.
>
> > I recommend that the authors substantially revise the paper to improve the mathematical clarity of the proposed methods and to further investigate why incorporating structural information does not yield meaningful performance gains.
>
> We thank the reviewer for this valuable feedback. In response to your suggestion, we have substantially revised the paper to improve the mathematical clarity and overall readability of the proposed methods. Concretely, we have
>
> - made the notation and definitions more explicit, e.g., for the message and update functions.
> - made the theorem statements more formal
> - provided more complete experimental details
> - revised the figures
> - reorganized the presentation of our framework
>
> Regarding the reviewer’s concern that adding structural information does not lead to significant performance gains, we note that our methods do improve the mean accuracy and often reduce variance. Since the TU benchmarks are highly saturated with strong baselines, large absolute improvements are naturally difficult to obtain, nevertheless, the gains we observe are consistent and practically meaningful. More importantly, our main contribution is to enhance the expressive power of MPNNs—particularly their ability to distinguish non-isomorphic graphs, while still achieving competitive performance on standard graph classification and regression tasks.
> > Q1: It is not clear why "kernel-based aggregation strategy effectively mitigating the discrepancies in node context distributions" leads to better performance.
>
> A1:We thank the reviewer for this question. In dense datasets such as DD, entropy values from different local structures can vary greatly in scale due to differing underlying distributions. Feeding these raw entropy values directly into an MPNN (as in the ConD variant) may cause large-magnitude entropies to dominate aggregation and hinder generalization. The KerA variant avoids this issue by comparing context distributions through a reproducing kernel, which maps entropy vectors into a bounded similarity space. This acts like a normalization step, converting absolute entropy differences into relative similarities and keeping node contributions on a comparable scale. Empirically, this kernel-based aggregation yields slightly better performance on several benchmarks. We have clarified this intuition in the revised manuscript (Section 5).
> > Q2:  In Figure 3 (a), does the entropy value depend on the central node?
>
> A2: In our method, an entropy value is computed **for each node** based on its own context subgraph. The “central node” in Figure 3(a) is only used for visualization purpose, we select one node and display the entropy values of its neighbors to illustrate the local distribution of entropies. Changing this central node for visualization would not change the entropy values themselves.
>
> [1] Yanbo Wang and Muhan Zhang. 2024. An empirical study of realized GNN expressiveness. In Proceedings of the 41st International Conference on Machine Learning (ICML'24), Vol. 235. JMLR.org, Article 2136, 52134–52155.

---

> > ### Comment · Reviewer_uqWp · 2025-11-27
> > **Response to Rebuttal**
> >
> > I thank the Reviewers for the rebuttal.
> >
> > While the method is interesting and provides better expressivity as shown in Table 1, the practical benefits are limited as in Table 3 and 9. While authors claim that the mean is higher, the standard deviation values are equally important to determine if the gain is statistically significant.
> >
> > **our intention in these Tables is not to argue that our message passing strategy leads to large performance jumps, but to demonstrate that the proposed plug-in frameworks (ConD- and KerA-) can be integrated into different backbones (GIN, K-GIN, GCN, and GraphSAGE) without hurting performance**
> >
> > --
> >
> > If the framework doesn't lead to significant improvement, then this questions the motivation behind having a new message passing framework. In this context, the following concern I had in the Review still remains:
> > > The current results directly challenge the core motivation behind injecting additional structural signals for representation learning: the results suggest that message passing schemes in standard architectures such as GCN and GraphSAGE may already implicitly capture the relevant structural information, and that explicitly adding more structure offers no tangible benefit in cases os benchmark or real-world graphs.
> >
> >
> > My assessment is that the paper remains below the acceptance threshold for a flagship venue. I encourage the authors to further investigate why incorporating structural information does not lead to meaningful performance gains on the benchmark datasets used, and to explore potential solutions. It is also possible that the proposed framework is more effective on graphs with particular structural properties rather than on general graphs. Clarifying this distinction would help better articulate the contribution.

---

### Meta-Review · Area_Chair_xGgf · 2026-01-07

**Summary:**

This paper explores augmenting message passing with entropy-based structural information from k-hop ego-subgraphs, which is an interesting and lightweight idea. However, reviewers raised serious concerns about unclear presentation, missing or confusing methodological details, and unsupported theoretical and efficiency claims. Empirically, performance gains are statistically insignificant or inconsistent, with missing comparisons to strong recent baselines. Overall, the experimental evidence and clarity do not substantiate the claimed benefits of the proposed method.

**Reviewer Concerns:**

Reviewers have concerns about unclear presentation, unsupported theoretical and efficiency claims, and missing comparisons to strong recent baselines.

**Reviewer Scores:**

Reviewers generally keep their scores after rebuttal.

---

### Decision · Program_Chairs · 2026-01-26

Reject